# Clinical usefulness of the pattern of non-adherence to anti-platelet regimen in stented patients (PARIS) thrombotic risk score to predict long-term all-cause mortality and heart failure hospitalization after percutaneous coronary intervention

**Joh Akama** [ORCID] *, **Takeshi Shimizu**, **Takuya Ando**, **Fumiya Anzai**, **Yuuki Muto**, **Yusuke Kimishima**, **Takatoyo Kiko**, **Akiomi Yoshihisa**, **Takayoshi Yamaki**, **Hiroyuki Kunii**, **Kazuhiko Nakazato**, **Takafumi Ishida**, **Yasuchika Takeishi**

Department of Cardiovascular Medicine, Fukushima Medical University, Fukushima, Japan

☯ These authors contributed equally to this work.
\* joeakama@fmu.ac.jp

## Abstract

### Background

The Patterns of non-Adherence to Anti-Platelet Regimen in Stented Patients (PARIS) thrombotic risk score has been proposed to estimate the risk of stent thrombotic events after percutaneous coronary intervention (PCI). However, the prognostic value of the PARIS thrombotic risk score for long term all-cause and cardiac mortalities, as well as hospitalization due to heart failure, has not yet been evaluated. Therefore, the aim of the present study was to evaluate the prognostic value of the PARIS thrombotic risk score for all-cause and cardiac mortalities and hospitalization due to heart failure following PCI.

### Methods and results

Consecutive 1,061 patients who underwent PCI were divided into three groups based on PARIS thrombotic risk score; low- (n = 320), intermediate- (n = 469) and high-risk (n = 272) groups. We followed up on all three groups for all-cause mortality, cardiac mortality and hospitalization due to heart failure. Kaplan-Meier analysis showed that all outcomes were highest in the high-risk group (P < 0.001, P = 0.022 and P < 0.001, respectively). Multivariate Cox proportional hazard analysis, adjusted for confounding factors, showed that the risk of all-cause mortality and hospitalization due to heart failure of the high-risk group were higher than those of the low-risk group (hazard ratios 1.76 and 2.14, P = 0.005 and P = 0.017, respectively).

### Conclusion

The PARIS thrombotic risk score is a significant prognostic indicator for all-cause mortality and hospitalization due to heart failure in patients after PCI.

**Data Availability Statement:** All relevant data are within the manuscript and its Supporting Information files.

**Funding:** The author(s) received no specific funding for this work.

**Competing interests:** The authors have declared that no competing interests exist.

## Introduction

Percutaneous coronary intervention (PCI) is a well-established therapy for patients suffering from coronary artery disease (CAD), and has developed into one of the most popular treatments in modern cardiovascular disease. A number of risk stratification tools or risk scores have been developed with the purpose of estimating prognosis after PCI over the years. Three well-known risk scores are the Global Registry of Acute Coronary Events (GRACE) score [1–3], the Thrombolysis in Myocardial Infarction (TIMI) score [4], and the Controlled Abciximab and Device Investigation to Lower Late Angioplasty Complications (CADILLAC) score [5]. These scores combine and weigh various predictors to calculate the risk of acute coronary syndrome (ACS) for an individual patient [6].

International guidelines focus on accurately predicting adverse events after PCIs such as stent thrombosis or bleeding events, because the magnitude of risk can aid in therapy selection and form the basis for a precise preprocedural informed consent practice [7–9]. Despite substantial improvements in PCIs with novel stent platforms and more potent P2Y12 inhibitors, coronary thrombotic events, such as myocardial infarction, stent thrombosis and revascularization, remain harmful complications with high rates of mortality [10].

The Pattern of non-Adherence to Anti-Platelet Regimen in Stented Patients (PARIS) thrombotic risk score [11] is a reliable tool for estimating the risks of stent thrombotic events in patients who have undergone PCI. Compared to the GRACE score, TIMI score and CADILLAC score, the advantage of the PARIS thrombotic risk score is that it is composed of simple risk scores of baseline clinical variables [11].

Death is an even more detrimental event than stent thrombosis as a potential complication following PCI. However, the clinical usefulness of the PARIS thrombotic risk score for long term all-cause and cardiac mortalities, as well as hospitalization due to heart failure, has not yet been evaluated. Therefore, we assessed the clinical usefulness of the PARIS thrombotic risk score for predicting long-term all-cause mortality, cardiac mortality and hospitalization due to heart failure in patients after undergoing PCI.

## Materials and methods

### Subjects and study protocol

This was an observational study, which enrolled a total of 1,061 consecutive patients who underwent their initial PCI due to stable angina pectoris or acute coronary syndrome (ACS) at Fukushima Medical University Hospital between January 2010 and May 2018. The exclusion criteria was defined as death at discharge. The diagnosis of stable angina pectoris and ACS was made by several cardiologists based on the definition in the cardiovascular disease guidelines [9, 12, 13]. The PARIS thrombotic risk score in the present study was calculated with six components including diabetes mellitus, ACS, current smoking, creatinine clearance < 60 ml/min, history of PCI and history of coronary artery bypass grafting (CABG), according to the original report [11]. We added 0–3 points depending on diabetes mellitus, 0–2 points depending on ACS, 0 or 1 points depending on current smoking, 2 points if creatinine clearance was less than 60 ml/min, and 2 points if the patients had a history of PCI or CABG. The range of integer scores was 0–12. We divided the patients into three groups based on their PARIS thrombotic risk score; low- (PARIS thrombotic risk score < 2, n = 320), intermediate- (PARIS thrombotic risk score 3–4, n = 469) and high-risk (PARIS thrombotic risk score ≥ 5, n = 272) groups. We compared the clinical features, laboratory data and parameters of echocardiography, as well as the characteristics of lesions, among the three groups.

The patients were followed up until April 2020 for all-cause death, cardiac death and hospitalization due to heart failure. All-cause death included cardiac death and death from respiratory failure, infection, sepsis, cancer, renal failure, stroke, digestive hemorrhage, and others. Cardiac death was confirmed by independent experienced cardiologists as death either from worsened heart failure, acute coronary syndrome, ventricular fibrillation documented by electrocardiograph or implantable devices, or sudden cardiac death. Status and dates of death were obtained from the patients' medical records, the attending physicians at the patients' referring hospital, the patients, or the patients' families via telephone. Physicians other than the attending physicians, who were blinded to the analyses of this study, conducted an investigation. Survival time was calculated from the date of PCI until the date of death or hospitalization due to heart failure. We were able to follow-up on all patients, and written informed consent was obtained from all of them. The study protocol was approved by the Ethics Committee of Fukushima Medical University (approved number 823), and was carried out in accordance with the principles outlined in the Declaration of Helsinki. Reporting of the study conforms to Strengthening the Reporting of Observational Studies in Epidemiology, along with references to Strengthening the Reporting of Observational Studies in Epidemiology and the broader Enhancing the Quality and Transparency of Health Research guidelines [14].

## Co-morbidities and previous history

Comorbidities were assessed by the patients' attending physicians. A smoker was defined as a patient with a current smoking habit or a habit that had been discontinued $\leq$ 6 months before hospitalization. Hypertension was defined as the recent use of antihypertensive drugs, a systolic blood pressure of $\geq$ 140 mmHg, and/or a diastolic pressure of $\geq$ 90 mmHg. Diabetes was defined as the recent use of insulin or antidiabetic drugs, a fasting blood glucose value of $\geq$ 126 mg/dL, and/or a hemoglobin A1c value of $\geq$ 6.5%. Dyslipidemia was defined as the recent use of cholesterol-lowering drugs, a triglyceride value of $\geq$ 150 mg/dL, a low-density lipoprotein (LDL) cholesterol value of $\geq$ 140 mg/dL, and/or a high-density lipoprotein (HDL) cholesterol value of $<$ 40 mg/dL. Chronic kidney disease (CKD) was defined as an estimated glomerular filtration rate (eGFR) of $<$ 60 mL/min/1.73 m$^2$ according to the Modification of Diet in Renal Disease formula [15, 16]. Anemia was defined as a hemoglobin level of $<$ 12.0 g/dL in females and $<$ 13.0 g/dL in males [17]. Atrial fibrillation was identified by an electrocardiogram performed during hospitalization and/or from medical records. The diagnosis of heart failure was made by several cardiologists based on the guidelines [17].

## Laboratory data and echocardiography

Blood samples were obtained at admission. Echocardiography was performed using standard techniques by experienced echocardiographers who were blinded to the present study. Left ventricular ejection fraction (LVEF) was calculated by using the modified Simpson's biplane method. All measurements were performed using ultrasound systems (ACUSON Sequoia, Siemens Medical Solutions USA, Inc, Mountain View, CA, USA).

## Lesions and therapies

Multivessel disease is defined as significant stenosis ($>$ 75%) in two or more major coronary arteries (of $\geq$ 2.5 mm diameter) [18]. Since the present study was an observational cohort study, a revascularization strategy of PCI and optimal medical therapy were selected at each physician's discretion. Unless serious bleeding complications occurred, the durations of dual anti-platelet therapy (DAPT) periods were three months after bare-metal stent implantation and one year after drug-eluting stent (DES) implantation, regardless of anticoagulation

therapy. After that, an aspirin or P2Y12 inhibitor monotherapy was continued. Triple therapy is defined as the prescription of both DAPT and oral anticoagulants.

## Statistical analysis

Statistical analysis was performed using SPSS version 25.0 (SPSS, Armonk, NY, USA) and EZR (Saitama Medical Center, Jichi Medical University, Saitama, Japan), which is a graphical user interface for R (The R Foundation for Statistical Computing, Vienna, Austria) [19]. In all analyses, a P value of < 0.05 was taken to indicate statistical significance. Continuous variables are presented as mean ± SD and compared by one-way ANOVA. The categorical variables are expressed as numbers and percentages, and the chi-square test was used for comparisons. The Kaplan-Meier method was used for presenting the mortality and the incidence of hospitalization due to heart failure, and statistical significance was analyzed using a log-rank test. The prognostic value of the PARIS bleeding risk score was tested by both univariate and multivariate Cox proportional hazard analyses. In the multivariate Cox proportional hazard analyses, the PARIS bleeding risk score was forcedly entered, and thereafter adjusted for clinical factors that were significantly different among the three groups in the univariate models, and were not included in the criteria of the PARIS bleeding risk score.

The cumulative incidence competing risk method was used for presenting the cardiac mortality and the incidence of hospitalization due to heart failure after adjusting competing risks such as non-cardiac death or all-cause death. To analyze differences in the cumulative incidence among the three groups, Gray's test was used. The model proposed by Fine and Gray was performed to reveal the hazard of the subdistribution and the relationship between covariates and cumulative incidence.

To compare the predictive ability for all-cause death over the first 6 months following PCI between PARIS thrombotic risk score and GRACE score, a receiver operating curve (ROC) analysis was performed for patients with ACS. The areas under the curve (AUC) of these two scores were compared using the DeLong test.

## Results

### Clinical characteristics

Of the 1,061 patients, 834 were male (78.6%) and had a mean age of 68.8 years. Comparisons of clinical characteristics in the low-, intermediate-, and high-risk groups are shown in Table 1. The high-risk group was significantly older and had a higher prevalence of histories of PCI, CABG, and old myocardial infarction. Additionally, the high-risk group had higher prevalence of insulin-treated diabetes mellitus, anemia, CKD, dialysis, peripheral artery disease and atrial fibrillation. Moreover, the high-risk group had a higher white blood cell count and hemoglobin A1c, and had lower hemoglobin, eGFR, albumin, LDL cholesterol and LVEF levels. Regarding medications, in the high-risk group, significantly more patients were prescribed β-blockers, renin-angiotensin-aldosterone system inhibitors and proton pump inhibitors compared to the other groups.

### Follow-up

During the follow-up period (mean 1,809 days), there were 205 all-cause deaths, including 64 cardiac deaths, and 100 hospitalizations due to heart failure. The 64 cardiac deaths included 48 deaths from heart failure, 10 deaths from acute myocardial infarction, and six deaths from ventricular arrhythmias. In the Kaplan-Meier analysis (Fig 1), all outcomes were highest in the high-risk group among the three groups (P < 0.001, P = 0.022 and P < 0.001, respectively).

**Table 1. Baseline characteristics of patients according to PARIS thrombotic risk score.**

| | Total | Low-risk group | Intermediate-risk group | High-risk group | P-value |
|---|---|---|---|---|---|
| | n = 1,061 | n = 320 | n = 469 | n = 272 | |
| PARIS thrombotic risk score | 3.4 ± 1.8 | 1.4 ± 0.7 | 3.4 ± 0.4 | 5.8 ± 1.0 | < 0.001 |
| Age, years | 68.8 ± 11.3 | 69.7 ± 10.9 | 67.4 ± 12.1 | 70.2 ± 10.2 | 0.001 |
| Male | 834 (78.6%) | 224 (70.0%) | 384 (82.5%) | 223 (82.0%) | < 0.001 |
| Body mass index, kg/m$^2$ | 24.2 ± 3.6 | 24.0 ± 3.6 | 24.2 ± 3.7 | 24.3 ± 3.6 | 0.545 |
| Smoking | 702 (66.7%) | 142 (45.7%) | 364 (77.6%) | 196 (72.1%) | < 0.001 |
| History of PCI | 180 (17.0%) | 12 (3.8%) | 66 (14.1%) | 102 (37.5%) | < 0.001 |
| History of CABG | 37 (3.5%) | 2 (0.6%) | 4 (0.9%) | 31 (11.4%) | < 0.001 |
| History of OMI | 89 (8.4%) | 19 (5.9%) | 28 (6.0%) | 42 (15.4%) | < 0.001 |
| Co-morbidities | | | | | |
| Hypertension | 902 (85.7%) | 261 (83.9%) | 402 (85.7%) | 239 (87.8%) | 0.397 |
| Diabetes mellitus | | | | | < 0.001 |
| Non-insulin-treated | 476 (44.9%) | 109 (34.1%) | 233 (49.7%) | 134 (49.3%) | |
| Insulin-treated | 65 (6.1%) | 0 (0%) | 10 (2.1%) | 55 (20.2%) | |
| Dyslipidemia | 972 (92.5%) | 286 (92.0%) | 435 (92.9%) | 251 (92.2%) | 0.828 |
| Anemia | 483 (45.9%) | 110 (35.4%) | 206 (43.9%) | 167 (61.4%) | < 0.001 |
| Chronic kidney disease | 461 (43.8%) | 27 (8.7%) | 194 (41.4%) | 240 (88.2%) | < 0.001 |
| Dialysis | 62 (5.9%) | 5 (1.6%) | 28 (6.0%) | 29 (10.7%) | < 0.001 |
| Peripheral artery disease | 132 (12.5%) | 31 (10.0%) | 46 (9.8%) | 55 (20.3%) | < 0.001 |
| Heart failure | 204 (19.2%) | 61 (19.1%) | 91 (19.4%) | 52 (19.1%) | 0.887 |
| Atrial fibrillation | 160 (15.2%) | 39 (12.5%) | 66 (14.1%) | 55 (20.2%) | 0.024 |
| Clinical presentations | | | | | < 0.001 |
| Stable angina pectoris | 535 (50.4%) | 241 (75.3%) | 208 (44.3%) | 86 (31.6%) | |
| STEMI | 316 (29.8%) | 34 (10.6%) | 164 (35.0%) | 118 (43.4%) | |
| NSTEMI | 91 (8.6%) | 6 (1.9%) | 45 (9.6%) | 40 (14.7%) | |
| Unstable angina pectoris | 119 (11.2%) | 39 (12.1%) | 52 (11.0%) | 28 (10.2%) | |
| Multivessel disease | 517 (48.7%) | 136 (42.5%) | 218 (46.5%) | 163 (59.9%) | < 0.001 |
| Target lesion | | | | | 0.001 |
| RCA | 340 (32.0%) | 89 (32.7%) | 169 (36.0%) | 82 (25.6%) | |
| LAD | 492 (46.4%) | 115 (42.3%) | 198 (42.2%) | 179 (55.9%) | |
| LCX | 189 (17.8%) | 50 (18.4%) | 88 (18.8%) | 51 (15.9%) | |
| LMT | 36 (3.4%) | 15 (5.5%) | 14 (3.0%) | 7 (2.2%) | |
| Graft | 3 (0.3%) | 3 (0.6%) | 0 | 0 | |
| Laboratory data | | | | | |
| White blood cell, × 10$^3$/μl | 8.2 ± 3.6 | 6.8 ± 2.4 | 8.4 ± 3.6 | 9.2 ± 4.2 | < 0.001 |
| Hemoglobin, g/dl | 13.2 ± 1.9 | 13.5 ± 1.6 | 13.3 ± 1.9 | 12.7 ± 2.2 | < 0.001 |
| eGFR, ml/min/1.73 m$^2$ | 64.0 ± 24.4 | 74.5 ± 16.4 | 65.8 ± 25.1 | 49.7 ± 23.7 | < 0.001 |
| CrCl, mL/min | 70.7 ± 34.4 | 79.1 ± 27.5 | 74.1 ± 37.4 | 54.9 ± 31.0 | < 0.001 |
| Albumin, g/dl | 3.8 ± 0.5 | 4.0 ± 0.4 | 3.8 ± 0.5 | 3.7 ± 0.5 | < 0.001 |
| LDL cholesterol, mg/dl | 105.5 ± 35.7 | 106.8 ± 35.4 | 108.0 ± 35.8 | 98.6 ± 35.5 | 0.037 |
| HDL cholesterol, mg/dl | 49.6 ± 18.8 | 51.7 ± 21.5 | 48.4 ± 16.0 | 47.9 ± 18.2 | 0.082 |
| Triglyceride, mg/dl | 139.1 ± 116.2 | 141.2 ± 85.4 | 140.7 ± 142.2 | 132.3 ± 109.4 | 0.741 |
| HbA1c, % | 6.1 ± 1.0 | 5.9 ± 0.8 | 6.1 ± 1.0 | 6.5 ± 1.3 | < 0.001 |
| Echocardiography | | | | | |
| LVEF, % | 56.3 ± 12.0 | 58.6 ± 10.4 | 55.0 ± 12.8 | 52.0 ± 12.6 | < 0.001 |
| Procedural characteristics | | | | | |
| BMS | 128 (12.1%) | 30 (9.4%) | 71 (15.1%) | 27 (9.9%) | 0.023 |

*(Continued)*

**Table 1.** (Continued)

|  | Total | Low-risk group | Intermediate-risk group | High-risk group | P-value |
|---|---|---|---|---|---|
|  | n = 1,061 | n = 320 | n = 469 | n = 272 |  |
| DES | 820 (77.3%) | 265 (82.8%) | 348 (74.2%) | 207 (76.1%) | 0.016 |
| DCB | 18 (1.7%) | 3 (0.9%) | 8 (1.7%) | 7 (2.5%) | 0.459 |
| POBA | 63 (5.9%) | 15 (4.7%) | 26 (5.5%) | 22 (8.1%) | 0.194 |
| Use of imaging device | 1031 (97.2%) | 313 (97.8%) | 458 (97.7%) | 260 (95.6%) | 0.149 |
| Use of physiologic guidance | 241 (22.7%) | 104 (32.5%) | 96 (20.5%) | 41 (15.1%) | < 0.001 |
| Medications |  |  |  |  |  |
| β-blockers | 670 (63.1%) | 154 (49.5%) | 310 (66.2%) | 206 (75.7%) | < 0.001 |
| RAS inhibitors | 797 (75.8%) | 193 (62.1%) | 382 (81.6%) | 222 (81.6%) | < 0.001 |
| Statins | 879 (83.6%) | 256 (80.0%) | 401 (85.5%) | 222 (81.6%) | 0.164 |
| Proton pump inhibitors | 792 (75.4%) | 218 (70.1%) | 353 (75.4%) | 221 (81.3%) | 0.008 |
| Triple therapy at discharge | 120 (11.3%) | 29 (9.0%) | 50 (10.6%) | 41 (15.0%) | 0.064 |

PARIS, the patterns of non-adherence to anti-platelet regimen in stented patients; PCI, percutaneous coronary intervention; CABG, coronary artery bypass grafting; OMI, old myocardial infarction; STEMI, ST elevation myocardial infarction; NSTEMI, non-ST elevation myocardial infarction; RCA, right coronary artery; LAD, left anterior descending coronary artery; LCX, left circumflex coronary artery LMT, left main trunk; eGFR, estimated glomerular filtration rate; CrCl, creatine clearance; LDL, low density lipoprotein; HDL, high density lipoprotein; HbA1c, hemoglobin A1c; LVEF, left ventricular ejection fraction; BMS, bare metal stent; DES, drug eluting stent; DCB, drug coated balloon; POBA, plain old balloon angioplasty; RAS, Renin-Angiotensin-Aldosterone System.

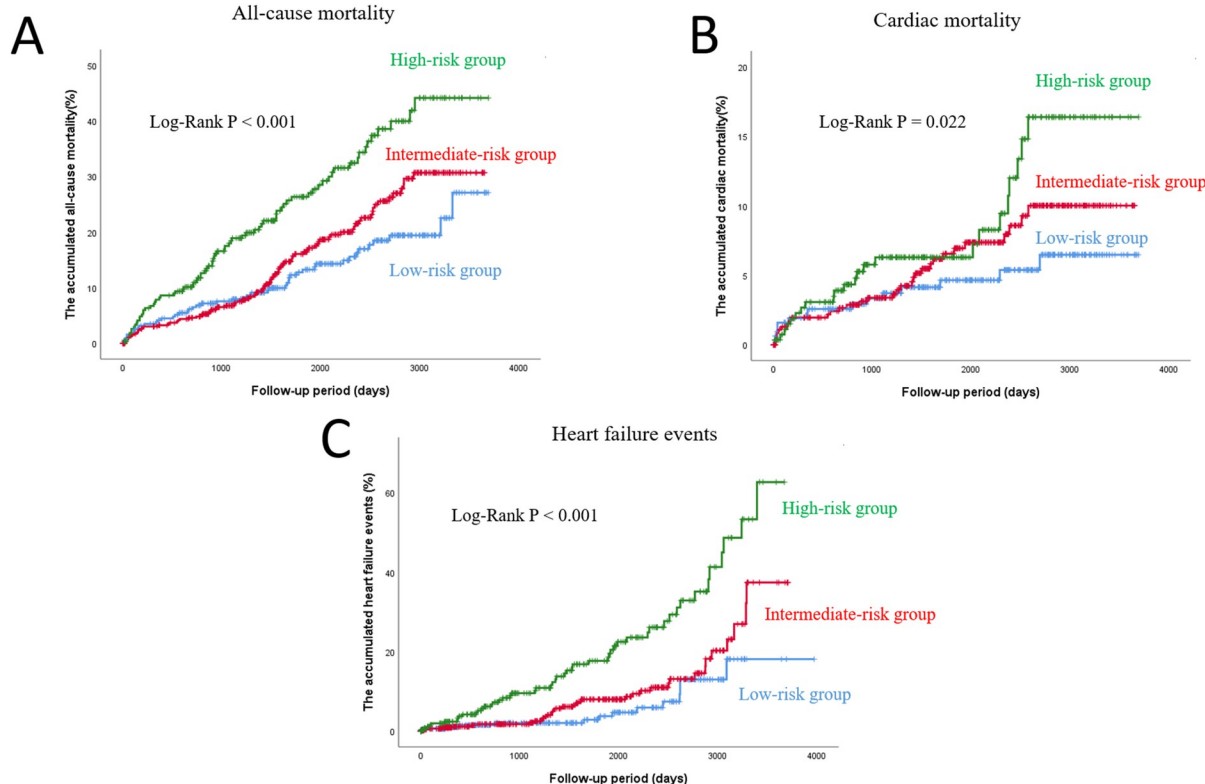

**Fig 1.** Kaplan-Meier curves for all-cause mortality (A), cardiac mortality (B) and hospitalization due to heart failure (C) in the high, intermediate, and low PARIS thrombotic risk score groups. PARIS, the Patterns of non-Adherence to anti-platelet Regimen in Stented patient.

Regarding cumulative incidence competing risk method (Fig 2), hospitalization due to heart failure after adjusting for competing risk was highest in high-risk group, compared to the other two groups (P < 0.001).

## Cox proportional hazard regression analysis

The Cox proportional hazard analysis of the associations between PARIS thrombotic score, presented as categorical variables (groups), with all-cause mortality, cardiac mortality, and hospitalization due to heart failure, are presented in Tables 2–4. The PARIS thrombotic risk score was revealed as an independent prognostic factor, and the Cox proportional hazard analysis demonstrated that the high-risk group had a higher risk of all-cause mortality and hospitalization due to heart failure compared with the low-risk group (hazard ratio 1.76, 95% confidence intervals 1.18–2.61, P = 0.005; hazard ratio 2.14, 95% confidence intervals 1.14–4.00, P = 0.017, respectively).

Furthermore, to assess the prognostic value of the PARIS thrombosis risk score for all-cause mortality, cardiac mortality, and incidence of hospitalization due to heart failure by clinical presentations, we conducted subgroup analysis (Fig 3). The incidence in the high-risk group was compared with that in the low-risk group in ACS subgroup. The hazard ratios of the PARIS thrombotic risk score on all-cause mortality in patients with stable angina pectoris and acute coronary syndrome were 1.72 (95% confidence intervals 1.01–2.93) and 3.30 (95% confidence intervals 1.63–6.67), respectively. The hazard ratios of the PARIS thrombotic risk score on cardiac mortality in patients presenting stable angina pectoris and acute coronary syndrome were 1.55 (95% confidence intervals 0.62–3.84) and 7.92 (95% confidence intervals 1.05–59.80), respectively. The hazard ratios of the PARIS thrombotic risk score on the incidence of hospitalization due to heart failure in patients presenting stable angina and acute coronary syndrome were 3.66 (95% confidence intervals 1.69–7.90) and 4.99 (95% confidence intervals 1.53–16.29), respectively. There were no interactions.

Moreover, we performed a Cox proportional hazard analysis for examining the association of the PARIS thrombotic risk score, presented as continuous variables (per 1-point increase),

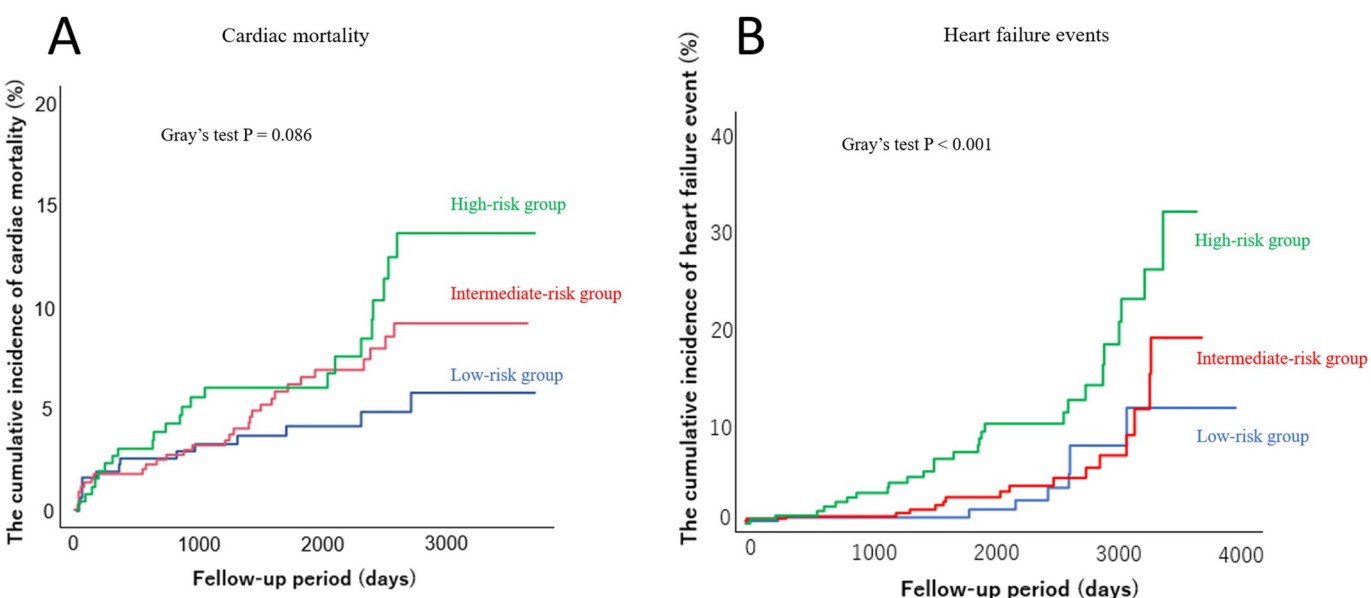

**Fig 2.** The cumulative incidence curves for cardiac mortality (A) and hospitalization due to heart failure (B) in the high, intermediate, and low PARIS thrombotic risk score groups. PARIS, the Patterns of non-Adherence to anti-platelet Regimen in Stented patient.

Table 2. Cox proportional hazard model of PARIS thrombotic score for all-cause mortality.

| Variable | Univariate analysis | | | Multivariate analysis[#] | | |
|---|---|---|---|---|---|---|
| | Hazard ratio | 95% Cl | P value | Hazard ratio | 95% Cl | P value |
| PARIS thrombotic score (low-risk group) | Ref. | | | Ref. | | |
| Intermediate-risk group (vs. low-risk group) | 1.32 | 0.92–1.88 | 0.120 | 1.42 | 0.97–2.07 | 0.066 |
| High-risk group (vs. low-risk group) | 2.34 | 1.62–3.37 | < 0.001 | 1.76 | 1.18–2.61 | 0.005 |
| Age (per 1-year increase) | 1.06 | 1.04–1.07 | < 0.001 | 1.04 | 1.03–1.06 | < 0.001 |
| Male | 0.90 | 0.70–1.36 | 0.902 | | | |
| Smoking | 0.97 | 0.72–1.29 | 0.851 | | | |
| History of PCI | 1.03 | 0.86–1.23 | 0.739 | | | |
| History of CABG | 0.70 | 0.52–0.95 | 0.026 | | | |
| History of OMI | 0.99 | 0.79–1.25 | 0.995 | | | |
| Diabetes mellitus | 1.13 | 0.86–1.49 | 0.359 | | | |
| Anemia | 3.58 | 2.62–4.89 | < 0.001 | 2.22 | 1.59–3.08 | < 0.001 |
| Chronic kidney disease | 2.65 | 1.99–3.52 | < 0.001 | | | |
| Dialysis | 3.45 | 2.32–5.11 | < 0.001 | | | |
| Peripheral artery disease | 1.76 | 1.24–2.50 | 0.001 | 1.22 | 0.85–1.75 | 0.272 |
| Atrial fibrillation | 2.06 | 1.50–2.84 | < 0.001 | 1.54 | 1.12–2.13 | 0.008 |
| Acute coronary syndrome | 1.14 | 0.86–1.50 | 0.339 | | | |
| Multivessel disease | 1.54 | 1.16–2.04 | 0.002 | 1.12 | 0.84–1.50 | 0.429 |
| Target lesion (RCA) | 0.98 | 0.73–1.31 | 0.912 | | | |
| Target lesion (LAD) | 0.15 | 0.62–1.07 | 0.818 | | | |
| Target lesion (LCX) | 1.24 | 0.88–1.74 | 0.210 | | | |
| Target lesion (LMT) | 1.52 | 0.78–2.97 | 0.216 | | | |
| LVEF (per 1% increase) | 0.97 | 0.95–0.98 | < 0.001 | 0.98 | 0.96–0.99 | < 0.001 |
| BMS | 0.95 | 0.65–1.37 | 0.795 | | | |
| DES | 0.88 | 0.65–1.20 | 0.442 | | | |
| β-blockers | 1.28 | 0.95–1.73 | 0.094 | | | |
| RAS inhibitors | 0.92 | 0.67–1.28 | 0.649 | | | |
| Proton pump inhibitor | 1.45 | 1.05–1.99 | 0.023 | 1.19 | 0.85–1.64 | 0.296 |

#. In the multivariate analysis, PARIS thrombotic score was forcedly entered, and thereafter adjusted for age, anemia, peripheral artery disease, atrial fibrillation, multivessel disease, LVEF and proton pump inhibitor with the forced entry method.

PARIS, the patterns of non-adherence to anti-platelet regimen in stented patients; PCI, percutaneous coronary intervention; CABG, coronary artery bypass grafting; OMI, old myocardial infarction; RCA, right coronary artery; LAD, left anterior descending coronary artery; LCX, left circumflex coronary artery LMT, left main trunk; LVEF, left ventricular ejection fraction; BMS, bare metal stent; DES, drug eluting stent; RAS, Renin-Angiotensin-Aldosterone System.

with all-cause and cardiac mortalities, and hospitalization due to heart failure as sensitivity analyses (Table 5). In the multivariable analysis, a high PARIS thrombotic score was determined to be an independent predictor of all-cause mortality and hospitalization due to heart failure after adjusting for other confounding factors (hazard ratio 1.12 per 1-point increase, 95% confidence intervals 1.04–1.20, P = 0.003; hazard ratio 1.21 per 1-point increase, 95% confidence intervals 1.09–1.35, P < 0.001, respectively).

## Fine and Gray model (subdistribution hazard model)

The Fine and Gray model (subdistribution hazard model) for the associations of the PARIS thrombotic score, presented as categorical variables (groups), with cardiac mortalities and hospitalization due to heart failure are presented in Tables 6 and 7.

**Table 3. Cox proportional hazard model of PARIS thrombotic score for cardiac mortality.**

| Variable | Univariate analysis | | | Multivariate analysis# | | |
|---|---|---|---|---|---|---|
| | Hazard ratio | 95% Cl | P value | Hazard ratio | 95% Cl | P value |
| PARIS thrombotic score (low-risk group) | Ref. | | | Ref. | | |
| Intermediate-risk group (vs. low-risk group) | 1.46 | 0.79–2.71 | 0.226 | 1.76 | 0.85–3.64 | 0.128 |
| High-risk group (vs. low-risk group) | 2.10 | 1.09–4.04 | 0.025 | 1.51 | 0.69–3.30 | 0.293 |
| Age (per 1-year increase) | 1.04 | 1.01–1.06 | 0.001 | 1.03 | 1.00–1.06 | 0.012 |
| Male | 1.07 | 0.58–1.97 | 0.814 | | | |
| Smoking | 1.16 | 0.68–1.99 | 0.579 | | | |
| History of PCI | 0.82 | 0.45–1.48 | 0.518 | | | |
| History of CABG | 0.67 | 0.40–1.11 | 0.124 | | | |
| History of OMI | 1.06 | 0.46–2.46 | 0.877 | | | |
| Diabetes mellitus | 1.17 | 0.71–1.92 | 0.524 | | | |
| Anemia | 3.20 | 1.86–5.53 | < 0.001 | 1.87 | 1.04–3.34 | 0.034 |
| Chronic kidney disease | 3.04 | 1.80–5.14 | < 0.001 | | | |
| Dialysis | 3.73 | 1.90–7.34 | < 0.001 | | | |
| Peripheral artery disease | 2.06 | 1.14–3.74 | 0.016 | 1.40 | 0.75–2.59 | 0.286 |
| Atrial fibrillation | 2.45 | 1.42–4.22 | 0.001 | 1.71 | 0.98–2.97 | 0.057 |
| Acute coronary syndrome | 0.92 | 0.56–1.51 | 0.754 | | | |
| Multivessel disease | 2.46 | 1.44–4.21 | 0.001 | 1.70 | 0.97–2.97 | 0.061 |
| Target lesion (RCA) | 0.65 | 0.37–1.12 | 0.127 | | | |
| Target lesion (LAD) | 0.93 | 0.57–1.49 | 0.764 | | | |
| Target lesion (LCX) | 1.63 | 0.94–2.83 | 0.077 | | | |
| Target lesion (LMT) | 1.99 | 0.72–5.47 | 0.181 | | | |
| LVEF (per 1% increase) | 0.94 | 0.93–0.96 | < 0.001 | 0.95 | 0.94–0.97 | < 0.001 |
| BMS | 0.32 | 0.11–0.89 | 0.029 | 0.37 | 0.13–1.04 | 0.060 |
| DES | 1.22 | 0.69–2.18 | 0.483 | | | |
| β-blockers | 2.02 | 1.12–3.66 | 0.019 | 1.39 | 0.74–2.63 | 0.301 |
| RAS inhibitors | 1.14 | 0.62–2.10 | 0.659 | | | |
| Proton pump inhibitor | 1.67 | 0.91–3.04 | 0.094 | | | |

#. In the multivariate analysis, PARIS thrombotic score was forcedly entered, and thereafter adjusted for age, anemia, peripheral artery disease, atrial fibrillation, multivessel disease, LVEF, BMS and β-blockers with the forced entry method.

PARIS, the patterns of non-adherence to anti-platelet regimen in stented patients; PCI, percutaneous coronary intervention; CABG, coronary artery bypass grafting; OMI, old myocardial infarction; RCA, right coronary artery; LAD, left anterior descending coronary artery; LCX, left circumflex coronary artery LMT, left main trunk; LVEF, left ventricular ejection fraction; BMS, bare metal stent; DES, drug eluting stent; RAS, Renin-Angiotensin-Aldosterone System.

Moreover, we performed the Fine and Gray model (subdistribution hazard model) of the associations of the PARIS thrombotic risk score, presented as continuous variables (per 1-point increase), with cardiac mortality and hospitalization due to heart failure as sensitivity analyses (Table 8). In the subdistribution multivariable models, a high PARIS thrombotic score was determined to be an independent predictor of hospitalization due to heart failure after adjusting for other confounding factors (hazard ratio 1.28 per 1-point increase, 95% confidence intervals 1.10–1.50, P = 0.001).

## ROC analysis

The AUC values of the PARIS thrombotic risk score and the GRACE score for predicting all-cause death in the first six months after PCI were 0.69 (95% confidence intervals 0.57–0.80)

**Table 4. Cox proportional hazard model of PARIS thrombotic score for hospitalization due to heart failure.**

| Variable | Univariate analysis | | | Multivariate analysis[#] | | |
|---|---|---|---|---|---|---|
| | Hazard ratio | 95% Cl | P value | Hazard ratio | 95% Cl | P value |
| PARIS thrombotic score (low-risk group) | Ref. | | | Ref. | | |
| Intermediate-risk group (vs. low-risk group) | 1.58 | 0.85–2.94 | 0.142 | 1.28 | 0.68–2.41 | 0.439 |
| High-risk group (vs. low-risk group) | 4.05 | 2.23–7.33 | < 0.001 | 2.14 | 1.14–4.00 | 0.017 |
| Age (per 1-year increase) | 1.05 | 1.03–1.07 | < 0.001 | 1.03 | 1.00–1.05 | 0.004 |
| Male | 0.68 | 0.44–1.05 | 0.087 | | | |
| Smoking | 0.73 | 0.49–1.09 | 0.133 | | | |
| History of PCI | 0.94 | 0.57–1.56 | 0.832 | | | |
| History of CABG | 0.33 | 0.16–0.69 | 0.003 | | | |
| History of OMI | 0.83 | 0.45–1.52 | 0.551 | | | |
| Diabetes mellitus | 1.43 | 0.95–2.15 | 0.084 | | | |
| Anemia | 3.56 | 2.28–5.58 | < 0.001 | 1.99 | 1.23–3.20 | 0.005 |
| Chronic kidney disease | 5.55 | 3.43–9.00 | < 0.001 | | | |
| Dialysis | 1.69 | 0.79–3.38 | 0.181 | | | |
| Peripheral artery disease | 2.24 | 1.39–3.61 | 0.001 | 1.26 | 0.75–2.09 | 0.372 |
| Atrial fibrillation | 3.10 | 2.06–4.69 | < 0.001 | 2.17 | 1.42–3.32 | < 0.001 |
| Acute coronary syndrome | 1.11 | 0.75–1.64 | 0.600 | | | |
| Multivessel disease | 1.66 | 1.10–2.50 | 0.015 | 1.07 | 0.69–1.66 | 0.742 |
| Target lesion (RCA) | 0.95 | 0.63–1.44 | 0.828 | | | |
| Target lesion (LAD) | 1.09 | 0.73–1.61 | 0.665 | | | |
| Target lesion (LCX) | 0.75 | 0.41–1.37 | 0.350 | | | |
| Target lesion (LMT) | 1.62 | 0.51–5.16 | 0.408 | | | |
| LVEF (per 1% increase) | 0.95 | 0.94–0.96 | < 0.001 | 0.96 | 0.95–0.98 | < 0.001 |
| BMS | 0.37 | 0.18–0.74 | 0.005 | 0.44 | 0.18–1.07 | 0.071 |
| DES | 1.71 | 1.05–2.78 | 0.030 | 0.97 | 0.51–1.81 | 0.923 |
| β-blockers | 2.01 | 1.23–3.29 | 0.005 | 1.21 | 0.71–2.05 | 0.478 |
| RAS inhibitors | 1.17 | 0.71–1.93 | 0.531 | | | |
| Proton pump inhibitor | 2.22 | 1.33–3.69 | 0.002 | 1.57 | 0.92–2.69 | 0.095 |

#. In the multivariate analysis, PARIS thrombotic score was forcedly entered, and thereafter adjusted for age, anemia, peripheral artery disease, atrial fibrillation, multivessel disease, LVEF, BMS, DES, β-blockers and proton pump inhibitor with the forced entry method.

PARIS, the patterns of non-adherence to anti-platelet regimen in stented patients; PCI, percutaneous coronary intervention; CABG, coronary artery bypass grafting; OMI, old myocardial infarction; RCA, right coronary artery; LAD, left anterior descending coronary artery; LCX, left circumflex coronary artery LMT, left main trunk; LVEF, left ventricular ejection fraction; BMS, bare metal stent; DES, drug eluting stent; RAS, Renin-Angiotensin-Aldosterone System.

and 0.68 (95% confidence intervals 0.52–0.84), respectively. There was no significant difference in AUC values between these two scores (Fig 4).

## Discussion

In the present study, of 1,061 patients who had undergone PCIs, we investigated the prognostic value of the PARIS thrombotic risk score for all-cause and cardiac mortalities, as well as hospitalization due to heart failure. As far as we have investigated, this study is the first to reveal the prognostic value of PARIS thrombotic score for long term all-cause mortality and hospitalization due to heart failure in the patients who had undergone PCIs, regardless of comorbidity severity, clinical presentation, or medication at discharge [20].

The risk stratification tools for coronary artery disease after PCIs such as the GRACE score [1–3], the TIMI risk score [4], and the CADILLAC score [5] were reported to have predictive

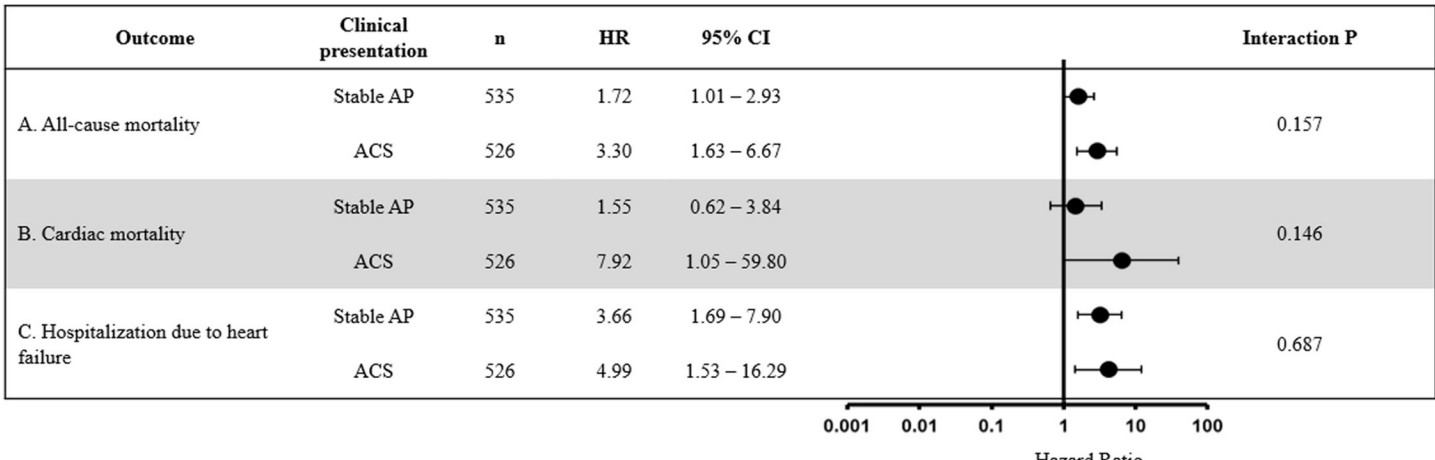

**Fig 3. Forest plot of hazard ratios by clinical presentations.** HR, hazard ratio; CI, confidence interval; AP, angina pectoris; ACS, acute coronary syndrome.

abilities for mortality in ACS patients after PCIs. However, there have been few risk scores that include patients with not only ACS, but also chronic coronary syndrome, and can be used to predict long term mortalities. The ability of PARIS thrombotic risk score to predict the 6-month mortality of patients with ACS was comparable to that of the GRACE score; hence, we examined the long-term prognostic value of PARIS thrombotic risk score for both ACS and chronic coronary syndrome. In addition, we have shown that the PARIS thrombotic risk score can predict events of heart failure hospitalization that the above scores do not. Heart failure remains a major cause of mortality, morbidity, hospitalization, and poor quality of life in patients with coronary artery disease [21]; thus, identifying patients at risk of heart failure is important to improve their prognosis after PCIs.

To date, some predictive scores for thrombotic events after PCIs also have been reported. The DAPT trial, TRA 2˚P-TIMI 50 (Thrombin Receptor Antagonist in Secondary Prevention of Atherothrombotic Ischemic Events–Thrombolysis in Myocardial Infarction 50) trial and CREDO-Kyoto thrombotic risk scores [22–24] have been developed to predict thrombotic events after PCIs. Compared to the thrombotic event risk predictive score mentioned above,

**Table 5. Cox proportional hazard model of PARIS thrombotic score (continuous variables) for all-cause and cardiac mortalities and hospitalization due to heart failure.**

| Variable | Univariate analysis | | | Multivariate analysis | | |
|---|---|---|---|---|---|---|
| | Hazard ratio | 95% Cl | P value | Hazard ratio | 95% Cl | P value |
| **All-cause mortality** | | | | | | |
| PARIS thrombotic score (per 1-point increase) [#1] | 1.18 | 1.10–1.27 | < 0.001 | 1.12 | 1.04–1.20 | 0.003 |
| **Cardiac mortality** | | | | | | |
| PARIS thrombotic score (per 1-point increase) [#2] | 1.18 | 1.04–1.34 | 0.007 | 1.09 | 0.96–1.25 | 0.173 |
| **Hospitalization due to heart failure** | | | | | | |
| PARIS thrombotic score (per 1-point increase) [#3] | 1.38 | 1.25–1.52 | < 0.001 | 1.21 | 1.09–1.35 | < 0.001 |

In the multivariate analysis, PARIS thrombotic score was adjusted for clinically important variables as shown below with the forced entry method.

#1: age, anemia, peripheral artery disease, atrial fibrillation, multivessel disease, LVEF and proton pump inhibitor

#2: age, anemia, peripheral artery disease, atrial fibrillation, multivessel disease, LVEF, BMS and β-blockers

#3: age, anemia, peripheral artery disease, atrial fibrillation, multivessel disease, LVEF, BMS, DES, β-blockers and proton pump inhibitor

PARIS, the patterns of non-adherence to anti-platelet regimen in stented patients; LVEF, left ventricular ejection fraction; BMS, bare metal stent; DES, drug eluting stent

**Table 6. Fine and Gray model (subdistribution hazard model) for cardiac mortality.**

| Variable | Univariate analysis | | | Multivariate analysis[#] | | |
|---|---|---|---|---|---|---|
| | sHR | 95% Cl | P value | sHR | 95% Cl | P value |
| PARIS thrombotic score (low-risk group) | Ref. | | | Ref. | | |
| Intermediate-risk group (vs. low-risk group) | 1.50 | 0.79–2.85 | 0.210 | 1.78 | 0.83–3.81 | 0.130 |
| High-risk group (vs. low-risk group) | 2.07 | 1.06–4.04 | 0.032 | 1.49 | 0.66–3.32 | 0.330 |
| Age (per 1-year increase) | 1.04 | 1.01–1.06 | 0.002 | 1.02 | 1.00–1.05 | 0.050 |
| Male | 0.95 | 0.53–1.71 | 0.890 | | | |
| Smoking | 1.23 | 0.71–2.13 | 0.460 | | | |
| History of PCI | 1.27 | 0.71–2.29 | 0.410 | | | |
| History of CABG | 2.44 | 0.98–6.06 | 0.054 | | | |
| History of OMI | 0.97 | 0.42–2.22 | 0.960 | | | |
| Diabetes mellitus | 1.16 | 0.70–1.93 | 0.540 | | | |
| Anemia | 3.01 | 1.72–5.27 | < 0.001 | 1.83 | 1.02–3.28 | 0.042 |
| Chronic kidney disease | 2.86 | 1.68–4.88 | < 0.001 | | | |
| Dialysis | 3.36 | 1.70–6.61 | < 0.001 | | | |
| Peripheral artery disease | 1.84 | 1.00–3.38 | 0.047 | 1.22 | 0.63–2.36 | 0.550 |
| Atrial fibrillation | 2.19 | 1.26–3.82 | 0.005 | 1.63 | 0.93–2.83 | 0.084 |
| Acute coronary syndrome | 0.83 | 0.51–1.34 | 0.450 | | | |
| Multivessel disease | 2.25 | 1.34–3.76 | 0.002 | 1.96 | 1.08–3.56 | 0.027 |
| Target lesion (RCA) | 0.62 | 0.35–1.09 | 0.098 | | | |
| Target lesion (LAD) | 0.93 | 0.57–1.51 | 0.800 | | | |
| Target lesion (LCX) | 1.67 | 0.96–2.91 | 0.065 | | | |
| Target lesion (LMT) | 2.04 | 0.73–5.70 | 0.170 | | | |
| LVEF (per 1% increase) | 0.94 | 0.93–0.96 | < 0.001 | 0.95 | 0.94–0.97 | < 0.001 |
| BMS | 0.33 | 0.12–0.95 | 0.041 | 0.40 | 0.13–1.20 | 0.100 |
| DES | 1.18 | 0.66–2.12 | 0.560 | | | |
| β-blockers | 2.36 | 1.26–4.42 | 0.007 | 1.69 | 0.86–3.30 | 0.130 |
| RAS inhibitors | 1.24 | 0.66–2.34 | 0.500 | | | |
| Proton pump inhibitor | 1.52 | 0.83–2.77 | 0.170 | | | |

#. In the multivariate analysis, PARIS thrombotic score was forcedly entered, and thereafter adjusted for age, anemia, peripheral artery disease, atrial fibrillation, multivessel disease, LVEF, BMS and β-blockers with the forced entry method.

sHR, subdistribution hazard ratio; PARIS, the patterns of non-adherence to anti-platelet regimen in stented patients; PCI, percutaneous coronary intervention; CABG, coronary artery bypass grafting; OMI, old myocardial infarction; RCA, right coronary artery; LAD, left anterior descending coronary artery; LCX, left circumflex coronary artery LMT, left main trunk; LVEF, left ventricular ejection fraction; BMS, bare metal stent; DES, drug eluting stent; RAS, Renin-Angiotensin-Aldosterone System.

the advantage of the PARIS thrombotic risk score is that its components only include patient characteristics without angiographical information. Therefore, it is easy to use in daily clinical practice. Present study has shown the possibility that physicians are able to apply the PARIS thrombotic score not only to estimate the risk of coronary thrombotic events, but also to assess the risk of death and worsening heart failure for patients after PCIs. The PARIS thrombotic risk score may be able to serve as a clinical tool for shared decision making to prevent thrombotic event, hospitalization due to heart failure and death in the clinical practice post PCI.

A few studies have reported on the association between PARIS thrombotic risk score and mortality in patients with PCIs. Zhao et al. reported that the PARIS thrombotic risk score was a significant independent predictor of 2-year mortality in patients after PCIs [25]. One of the advantages of the present study is that our follow-up period was much longer than that in

**Table 7. Fine and Gray model (subdistribution hazard model) for hospitalization due to heart failure.**

| Variable | Univariate analysis | | | Multivariate analysis[#] | | |
|---|---|---|---|---|---|---|
| | sHR | 95% Cl | *P* value | sHR | 95% Cl | *P* value |
| PARIS thrombotic score (low-risk group) | Ref. | | | Ref. | | |
| Intermediate-risk group (vs. low-risk group) | 1.18 | 0.51–2.72 | 0.700 | 0.99 | 0.42–2.32 | 0.980 |
| High-risk group (vs. low-risk group) | 3.23 | 1.47–7.09 | 0.003 | 2.10 | 0.89–4.95 | 0.088 |
| Age (per 1-year increase) | 1.01 | 0.98–1.04 | 0.330 | | | |
| Male | 0.52 | 0.29–0.91 | 0.022 | 0.55 | 0.30–1.00 | 0.051 |
| Smoking | 0.82 | 0.47–1.44 | 0.500 | | | |
| History of PCI | 1.40 | 0.75–2.60 | 0.280 | | | |
| History of CABG | 3.16 | 1.31–7.63 | 0.010 | | | |
| History of OMI | 1.38 | 0.66–2.89 | 0.380 | | | |
| Diabetes mellitus | 1.78 | 0.99–3.20 | 0.054 | | | |
| Anemia | 2.28 | 1.25–4.16 | 0.006 | 1.40 | 0.74–2.64 | 0.300 |
| Chronic kidney disease | 3.85 | 2.02–7.33 | < 0.001 | | | |
| Dialysis | 0.80 | 0.27–2.37 | 0.690 | | | |
| Peripheral artery disease | 2.06 | 1.09–3.89 | 0.025 | 1.23 | 0.62–2.41 | 0.550 |
| Atrial fibrillation | 3.12 | 1.82–5.35 | < 0.001 | 2.37 | 1.34–4.19 | 0.003 |
| Acute coronary syndrome | 1.07 | 0.62–1.85 | 0.780 | | | |
| Multivessel disease | 1.57 | 0.89–2.79 | 0.120 | | | |
| Target lesion (RCA) | 0.92 | 0.51–1.65 | 0.800 | | | |
| Target lesion (LAD) | 1.24 | 0.72–2.14 | 0.430 | | | |
| Target lesion (LCX) | 0.56 | 0.22–1.38 | 0.210 | | | |
| Target lesion (LMT) | 1.77 | 0.42–7.34 | 0.430 | | | |
| LVEF (per 1% increase) | 0.96 | 0.95–0.96 | < 0.001 | 0.98 | 0.96–0.99 | 0.044 |
| BMS | 0.31 | 0.11–0.84 | 0.021 | 0.66 | 0.18–2.38 | 0.530 |
| DES | 2.18 | 1.10–4.34 | 0.026 | 1.69 | 0.66–4.37 | 0.270 |
| β-blockers | 2.28 | 1.10–4.72 | 0.026 | 1.48 | 0.70–3.13 | 0.300 |
| RAS inhibitors | 1.33 | 0.65–2.73 | 0.420 | | | |
| Proton pump inhibitor | 2.05 | 1.06–3.98 | 0.032 | 1.32 | 0.69–2.52 | 0.390 |

#. In the multivariate analysis, PARIS thrombotic score was forcedly entered, and thereafter adjusted for male, anemia, peripheral artery disease, atrial fibrillation, LVEF, BMS, DES, β-blockers and proton pump inhibitor with the forced entry method.

sHR, subdistribution hazard ratio; PARIS, the patterns of non-adherence to anti-platelet regimen in stented patients; PCI, percutaneous coronary intervention; CABG, coronary artery bypass grafting; OMI, old myocardial infarction; RCA, right coronary artery; LAD, left anterior descending coronary artery; LCX, left circumflex coronary artery LMT, left main trunk; LVEF, left ventricular ejection fraction; BMS, bare metal stent; DES, drug eluting stent; RAS, Renin-Angiotensin-Aldosterone System.

Zhao et al.'s report. Their study population only included patients who had undergone drug eluting stent (DES) implantation, whereas the present study also included patients who had not undergone such implantation, which is useful in terms of stent-less PCI. Moreover, their study did not include a multivariate analysis in assessing the prognostic value of the PARIS thrombotic risk score for all-cause mortality. In contrast, in the present study we performed multivariate Cox proportional hazard analysis followed by competing risk analysis to assess the prognostic value of the PARIS thrombotic risk score. Therefore, the present study suggests that it may provide a new insight into the clinical usefulness of the PARIS thrombotic risk score.

The present study had several limitations. First, as a retrospective cohort study, the study may have inherent limitations regarding selection and ascertainment biases. Second, the

**Table 8. Fine and Gray model (subdistribution hazard model) of PARIS thrombotic score (continuous variables) for cardiac mortality and hospitalization due to heart failure.**

| Variable | Univariate analysis | | | Multivariate analysis | | |
|---|---|---|---|---|---|---|
| | sHR | 95% Cl | *P* value | sHR | 95% Cl | *P* value |
| **Cardiac mortality** | | | | | | |
| PARIS thrombotic score (per 1-point increase) [#1] | 1.17 | 1.03–1.33 | 0.014 | 1.08 | 0.95–1.23 | 0.210 |
| **Hospitalization due to heart failure** | | | | | | |
| PARIS thrombotic score (per 1-point increase) [#2] | 1.36 | 1.19–1.56 | < 0.001 | 1.28 | 1.10–1.50 | 0.001 |

In the multivariate analysis, PARIS thrombotic score was adjusted for clinically important variables as shown below with the forced entry method.

#1. age, anemia, peripheral artery disease, atrial fibrillation, multivessel disease, LVEF, BMS and β-blockers

#2. male, anemia, peripheral artery disease, atrial fibrillation, LVEF, BMS, DES, β-blockers and proton pump inhibitor

PARIS, the patterns of non-adherence to anti-platelet regimen in stented patients; sHR, subdistribution hazard ration; LVEF, left ventricular ejection fraction; BMS, bare metal stent; DES, drug eluting stent

clinical practices regarding medical treatment and interventional treatment in the present study may have some difference in the setting of each period. Especially, the achievement rate of guideline directed medical therapy for coronary artery disease and the presence or absence of interventional treatment may affect the rate of thrombotic events, death and heart failure hospitalization after PCI. Third, it may be difficult to determine the reproducibility and generalizability of PARIS thrombotic risk score to new patient or different outcome because the

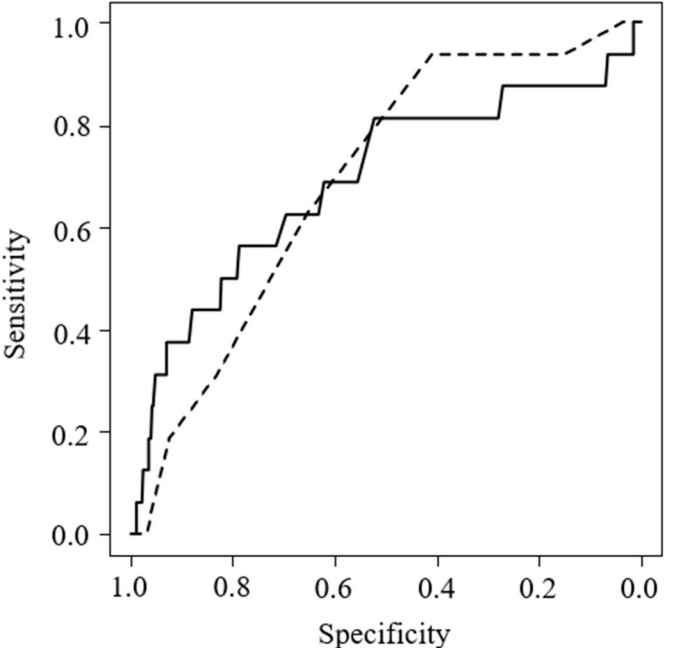

| | AUC | 95% CI | P-value |
|---|---|---|---|
| The PARIS thrombotic risk score | 0.69 | 0.57 - 0.80 | Reference |
| The GRACE score | 0.68 | 0.52 - 0.84 | 0.978 |

**Fig 4. Receiving operating curves (ROC) to predict mortality in 6 months in patients with acute coronary syndrome.** PARIS, the Patterns of non-Adherence to anti-platelet Regimen in Stented patient; GRACE, Global Registry of Acute Coronary Events; AUC, area under the curve; CI, confidence interval.

results of the present study have yet to undergo external validation Fourth, we used only variables with regard to hospitalization in this study, without taking into consideration changes in medical parameters or post-discharge treatment.

## Conclusions

In patients who underwent PCIs, a high PARIS thrombotic risk score was associated with an increased risk of long-term all-cause death and hospitalization due to heart failure. Therefore, the PARIS thrombotic risk score could be useful not only for the risk stratification of thrombotic events after PCIs, but also for predicting all-cause mortality and hospitalization due to heart failure in patients who have undergone PCIs.

## Supporting information

**S1 File.**
(XLSX)

## Author Contributions

**Formal analysis:** Joh Akama, Takeshi Shimizu, Akiomi Yoshihisa.

**Investigation:** Joh Akama, Takeshi Shimizu, Takuya Ando, Fumiya Anzai, Yuuki Muto, Yusuke Kimishima, Takatoyo Kiko, Takayoshi Yamaki, Hiroyuki Kunii, Kazuhiko Nakazato, Takafumi Ishida.

**Methodology:** Joh Akama, Takeshi Shimizu, Akiomi Yoshihisa, Takafumi Ishida, Yasuchika Takeishi.

**Supervision:** Yasuchika Takeishi.

**Writing – original draft:** Joh Akama, Takeshi Shimizu.

**Writing – review & editing:** Yasuchika Takeishi.

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
