## [Decision Letter · Decision Letter 0]

24 Jan 2022

PONE-D-21-38539Clinical usefulness of the pattern of non-adherence to anti-platelet regimen in stented patients (PARIS) thrombotic risk score to predict long-term all-cause mortality and heart failure hospitalization after percutaneous coronary intervention.PLOS ONE

Dear Dr. Akama,

Thank you for submitting your manuscript to PLOS ONE. After careful consideration, we feel that it has merit but does not fully meet PLOS ONE’s publication criteria as it currently stands. Therefore, we invite you to submit a revised version of the manuscript that addresses the points raised during the review process. In particular, the authors need to make a compelling argument for why the PARIS score, which was designed to asses risk of thrombotic complications after PCI, should be used to assess mortality and heart failure risk over other risk scores that were designed for this purpose. The impact of the manuscript would be improved with a formal comparison with other appropriate risk scores. 

We look forward to receiving your revised manuscript.

Kind regards,

Jeffrey J. Rade, MD

Academic Editor

PLOS ONE

Journal Requirements:

- Ando T, Nakazato K, Kimishima Y, Kiko T, Shimizu T, Misaka T, Yamada S, Kaneshiro T, Yoshihisa A, Yamaki T, Kunii H, Takeishi Y. The clinical value of the PRECISE-DAPT score in predicting long-term prognosis in patients with acute myocardial infarction. Int J Cardiol Heart Vasc. 2020 Jun 7;29:100552. doi: 10.1016/j.ijcha.2020.100552. PMID: 32551359; PMCID: PMC7287192.

This is regarding lines 259-263 in the Discussion section of your manuscript.

- Takuya Ando, Kazuhiko Nakazato, Yusuke Kimishima, Takatoyo Kiko, Takeshi Shimizu, Tomofumi Misaka, Shinya Yamada, Takashi Kaneshiro, Akiomi Yoshihisa, Takayoshi Yamaki, Hiroyuki Kunii, Yasuchika Takeishi, The clinical value of the PRECISE-DAPT score in predicting long-term prognosis in patients with acute myocardial infarction, IJC Heart & Vasculature, Volume 29, 2020, 100552, ISSN 2352-9067, https://doi.org/10.1016/j.ijcha.2020.100552.

This is regarding lines 311 – 319 in the Discussion and Conclusion section of your manuscript.

In your revision ensure you cite all your sources (including your own works), and quote or rephrase any duplicated text outside the methods section. Further consideration is dependent on these concerns being addressed.

Reviewers' comments:

Reviewer's Responses to Questions

**Comments to the Author**

1. Is the manuscript technically sound, and do the data support the conclusions?

Reviewer #1: Yes

Reviewer #2: Partly

2. Has the statistical analysis been performed appropriately and rigorously? 

Reviewer #1: Yes

Reviewer #2: Yes

3. Have the authors made all data underlying the findings in their manuscript fully available?

Reviewer #1: Yes

Reviewer #2: No

4. Is the manuscript presented in an intelligible fashion and written in standard English?

Reviewer #1: Yes

Reviewer #2: No

5. Review Comments to the Author

Reviewer #1: We congratulate the authors on a rigorous prospective cohort study albeit single center and observational. Completing the mammoth undertaking of long-term follow up of these patients is truly an accomplishment in itself. Clear definitions were provided of the parameters established. Looking at a thrombotic risk score having predictive value in CHF hospitalization has novel application and using a score that does not apply angiographic information has utility even to the general practitioner of medicine or cardiology. Noting that the study includes long-term follow up is key and is hat sets this apart.

That being said however the study is underpowered being that it is single centered and with few patients. Also as the authors have pointed out that nearly 80% of it study subjects are men. And being that these patients are from 2010-2018, there are certain caveats when applying any risk score that does take into account angiographic findings such as how the PCIs were done i.e. use of imaging and physiologic guidance, stent variety and generation, taking into account the usage of DCBs which are not applicable to coronaries in the US, the ~10% use of BMS, etc. Also of note and as mentioned by the authors in the lines 312 -315 variables only related to the hospitalization were taken into account for. Thus the large part of our therapy outside of PCI as related to medication optimization with goal directed dosage levels was not addressed with this study. The tables were extensive and had multiple different characteristics that were compiled. However, these issues we believe are minor and more critique of the content rather than any major issues.

Minor Issues are grammatical in lines 48-53 would suggest to better copyedit this part "...and has developed (into) one of the most popular treatment(s) in modern cardiovascular disease. It is necessary for (would take out "the") patients who under(go) PCI(no plural s required)..." Lastly also a minor issue in our opinion there is a lack of discussion regarding the findings of cardiac death in the Low, Intermediate, High Risk PARIS score patients as described in Graph B of Figure 1. The focus as it rightly should be on all-cause mortality and CHF events. It would be ideal and interesting to discuss the findings of cardiac death in particular perhaps why these grafts initially partial converge and then after one sees the high risk patients diverge from the intermediate and low risk patients around the ~800 days mark.

Again we congratulate the authors on a well done study with great long-term follow up.

Reviewer #2: The paper presented by Akama et al. on the prognostic utility of the PARIS score on outcomes post PCI is a decently done study with a good message and design. The authors show the PARIS score commonly used to predict thrombotic complications post PCI can also be used to predict long-term mortality and heart failure hospitalizations. However, authors need to address some major limitations as described below before to substantially improve the quality of their work and the message delivered:

1- Study rationale and hypothesis: The authors chose to calculate the PARIS score for all subjects and present a study to justify its prognostic utility for long term outcomes. While the study population, follow-up duration, comprehensiveness of data and event rate is suitable for such goal, the choice of PARIS score over other available and widely used score for ACS and PCI such as GRACE, TIMI, CADILLAC, among others is not well justified in the introduction or the discussion. The authors highlighted the strengths of their study in comparison to those done by others in the discussion, but it is not true that GRACE and TIMI were not studied for longer term outcomes. Further, the PARIS score albeit simple and easy to use yet is limited with choice of variables (no age, gender, and co-morbidities such as anemia included in the score). Hence, in order to give validity to their study, I most certainly recommend comparing the prognostic value of PARIS score in this population against GRACE and TIMI score and see if it outperforms either one. By doing so, you will give your study more clinical meaning and strength to your chosen score.

2- Study design and methodology: This is a retrospective cohort study not a prospective one as highlighted in the limitations section. The authors nicely outlined all the steps they undertook to ascertain their outcomes and all the clinical variables of the study, all of which are strengths that can be highlighted in the limitation section.

Recommendations:

- However, there has to be data provided on (1) the type of ACS (STEMI, NSTEMI, UA) and (2) any exclusion criteria (did you include consecutive patients, exclude those with unavailable data on follow-up or STEMI, etc.).

- Give the presence of competing risk between HF hospitalization and cardiac mortality, the authors are urged to present competing risks survival analysis in their Cox regression for the HF hospitalization and cardiac mortality outcomes.

- I recommend to do a stratified analysis on the basis on presentation (ACS vs stable angina) as it is important to see if the PARIS score's prognostic potential holds for either presentation. Comparing the score with GRACE and TIMI is invalid for those with stable angina.

3- Limitations: The biggest limitation of the study is not the small sample size as authors mentioned but (1) retrospective design with inherent limitation to selection and ascertainment biases, (2) the absence of external validation of the proposed score, (3) absence of comparison of PARIS score with other widely used and validated risk scores such as GRACE and TIMI.

4- Language, technical terms and design: The authors need to improve the writing of the paper in terms of English language, particularly in the introduction. The words "dependable" and "helpful" to describe PCI are not appropriate, and repeating the aims statement with the previous sentence of what's missing in the literature (i.e. the long-term prognostic use of PARIS score) is also not needed. I suggest getting the help of a native English language speaker to proofread the paper.

Further, I could not fully read the tables presented as they were embedded in the text and truncated on the right, making half the tables missing to me. The KM curves are also of very poor quality and need to be presented in a higher resolution.

6. PLOS authors have the option to publish the peer review history of their article (what does this mean?). If published, this will include your full peer review and any attached files.

Reviewer #1: No

Reviewer #2: No

---

## [Author Response · Author response to Decision Letter 0]

3 Mar 2022

Responses to Academic Editor‘s comment

Editor’s comment

In particular, the authors need to make a compelling argument for why the PARIS score, which was designed to asses risk of thrombotic complications after PCI, should be used to assess mortality and heart failure risk over other risk scores that were designed for this purpose. The impact of the manuscript would be improved with a formal comparison with other appropriate risk scores.

Auther’s Reply

Thank you for your helpful comment. We received a similar comment from Reviewer#2. As you pointed out, the descriptions of conventional risk scores to estimate mortality including GRACE, TIMI and CADILLAC were lacking in our original manuscript. According to your suggestion, we have revised the Introduction that firstly described the characteristics of those well-known risk scores, which could estimate the short- to mid-term outcomes of ACS patients (GRACE predicts 6 months mortality, TIMI predicts 14 days, and CADILLAC predicts 1 month and 1-year mortalities). To assess the prognostic ability of PARIS thrombotic risk score for predicting mortality, we additionally conducted the receiver operating curve analysis for PARIS thrombotic risk score and GRACE risk score in patients with ACS and compared the areas under the curves (AUC) between two scores. The AUC for predicting 6 months mortality was 0.69 in PARIS thrombotic risk score, which was comparable to 0.68 in GRACE score, as described below. The equivalence of PARIS thrombotic score to GRACE score for predicting mid-term mortality gives validity to the study, therefore we highlighted the long-term prognostic value of PARIS thrombotic risk score for not only all-cause mortality but also cardiac mortality and heart failure event, in addition to the original role for prediction of thrombotic event.

We have added the result of ROC analysis in the revised manuscript (page　34, line 336 – 339 and Fig. 4), and described about GRACE, TIMI and CADILLAC in Introduction (page 3, line55 – 60) and Discussion (page 34 – 35, line 348 – 373). 

“The AUC values of the PARIS thrombotic risk score and the GRACE score for all-cause mortality in 6 months were 0.69 (95% confidence intervals 0.57–0.80) and 0.68 (95% confidence intervals 0.52–0.84). ROC analysis demonstrated that there was no significant difference between these two scores (Fig. 4).” (page 34, line 336 - 339)

Figure 4. ROC to predict all-cause mortality for 6 months in patients with acute coronary syndrome (ACS). 

“Three well-known risk scores are the Global Registry of Acute Coronary Events (GRACE) score [1-3], the Thrombolysis in Myocardial Infarction (TIMI) score[4] and the Controlled Abciximab and Device Investigation to Lower Late Angioplasty Complications (CADILLIAC) score[5]. These scores combine and weigh various predictors to calculate the risk of acute coronary syndrome (ACS) for an individual patient[6].” (page 3, line55 - 60)

“The risk stratification tools for coronary artery disease after PCIs such as the GRACE score[1-3] ,the TIMI[4] and the CADILLIAC score[5] were reported to have predictive abilities for mortality in patients after PCIs. The GRACE score, which is used for overall risk assessment in patients with ACS, including ST elevation myocardial infarction (STEMI) and non-ST elevation ACS (NSTE-ACS) was designed to calculate the probability of death or myocardial infarction during admission and at 6 months by weighting 8 risk factors: age, heart rate, systolic blood pressure, initial serum creatine, Killip class, hospitalization due to cardiac arrest, positive cardiac biomarker, and ST-segment deviation[1-3, 22]. The TIMI risk score, which is often used for risk assessment in patients with NSTE-ACS, is calculated by using 7 factors: age (≥65 years); at least 3 coronary risk factors (family history, hypertension, dyslipidemia, diabetes mellitus, and smoking); known significant (≥50%) coronary stenosis; ST changes ≥ 0.5 mm on electrocardiogram; at least 2 episodes of angina in 24 hours; aspirin use in the past 7 days; and increased cardiac markers. Most of these factors can be assessed immediately after the admission. As the score increases, the incidence of major cardiovascular complications in the following 2 weeks increases synergistically[4, 22]. The CADILLIAC score, which is often used for risk assessment in patients with STEMI, evaluates several different prognostic variables including LVEF, creatinine clearance, Killip class, final TIMI flow, age, anemia and presence of three vessel disease. Utilizing the CADILLIAC risk score patients were stratified by low (CADILLIAC score 0-2), intermediate (CADILLIAC score 3-5), and high (CADILLIAC score ≥ 6) risk groups. The CADILLIAC score accurately predict 30-day and 1-year mortality after primary PCI[5].” (page 34 - 35, line 348 - 373)

We have cited the following papers as new references (#1-6, 22).

[1] Granger CB, Goldberg RJ, Dabbous O, Pieper KS, Eagle KA, Cannon CP, et al. Predictors of hospital mortality in the global registry of acute coronary events. Arch Intern Med. 2003;163:2345-53.

[2] Eagle KA, Lim MJ, Dabbous OH, Pieper KS, Goldberg RJ, Van de Werf F, et al. A validated prediction model for all forms of acute coronary syndrome: estimating the risk of 6-month postdischarge death in an international registry. Jama. 2004;291:2727-33.

[3] Fox KA, Dabbous OH, Goldberg RJ, Pieper KS, Eagle KA, Van de Werf F, et al. Prediction of risk of death and myocardial infarction in the six months after presentation with acute coronary syndrome: prospective multinational observational study (GRACE). Bmj. 2006;333:1091.

[4] Antman EM, Cohen M, Bernink PJ, McCabe CH, Horacek T, Papuchis G, et al. The TIMI risk score for unstable angina/non-ST elevation MI: A method for prognostication and therapeutic decision making. Jama. 2000;284:835-42.

[5] Halkin A, Singh M, Nikolsky E, Grines CL, Tcheng JE, Garcia E, et al. Prediction of mortality after primary percutaneous coronary intervention for acute myocardial infarction: the CADILLAC risk score. J Am Coll Cardiol. 2005;45:1397-405.

[6] Poldervaart JM, Langedijk M, Backus BE, Dekker IMC, Six AJ, Doevendans PA, et al. Comparison of the GRACE, HEART and TIMI score to predict major adverse cardiac events in chest pain patients at the emergency department. Int J Cardiol. 2017;227:656-61.

[22] Kimura K, Kimura T, Ishihara M, Nakagawa Y, Nakao K, Miyauchi K, et al. JCS 2018 Guideline on Diagnosis and Treatment of Acute Coronary Syndrome. Circ J. 2019;83:1085-196.

 

Responses to Reviewer #1

We appreciate your detailed reviews on our manuscript. We have revised our manuscript according to your comments. 

Reviewer’s comment

That being said however the study is underpowered being that it is single centered and with few patients. Also as the authors have pointed out that nearly 80% of it study subjects are men. And being that these patients are from 2010-2018, there are certain caveats when applying any risk score that does take into account angiographic findings such as how the PCIs were done i.e. use of imaging and physiologic guidance, stent variety and generation, taking into account the usage of DCBs which are not applicable to coronaries in the US, the ~10% use of BMS, etc. Also of note and as mentioned by the authors in the lines 312 -315 variables only related to the hospitalization were taken into account for. Thus the large part of our therapy outside of PCI as related to medication optimization with goal directed dosage levels was not addressed with this study. The tables were extensive and had multiple different characteristics that were compiled. However, these issues we believe are minor and more critique of the content rather than any major issues.

Author’s Reply

Thank you for your important comments. As you mentioned, 78.6% of patients in our study are men, which may reflect the true prevalence in Japanese population of patients with coronary artery disease. The other large registries of coronary artery disease patients who underwent percutaneous coronary intervention (PCI) in Japan showed that 78.3% were male in PENDULUM registry [Nakamura M, Kadota K, Nakao K, Nakagawa Y, Shite J, Yokoi H, et al. High bleeding risk and clinical outcomes in East Asian patients undergoing percutaneous coronary intervention: the PENDULUM registry. EuroIntervention 2021; 19: 1154-1162] and 73.0% were male in CREDO-Kyoto registry [Application of the Modified High Bleeding Risk Criteria for Japanese Patients in an All-Comers Registry of Percutaneous Coronary Intervention -From the CREDO-Kyoto Registry Cohort-3-. Circ J 2021; 85: 769-781]. The duration of enrollment in this study was relatively long, from 2010 to 2018, thus there might be some differences in strategies for treatment, as you mentioned. Hence, we have analyzed the proportion of use of imaging devices during PCI, which was over 95% in all three groups as additionally displayed in Table 1, so that showing the consistent imaging device-guided PCI was performed during all periods. However, unfortunately, data of stent generation and goal directed dosage levels of medication were not available, though those were very important information for understanding the patient background and relationship to the outcomes, as you pointed out. 

Therefore, we have corrected the limitations as follow, reflecting your comments and also suggestions from Reviewer #2.

“The present study had several limitations. First, as a retrospective cohort study, the study may be with inherent limitation to selection and ascertainment biases. Second, clinical practices in PCI and medical treatment in the study may have some differences in the setting of each period. Third, it may be difficult to determine the reproducibility and generalizability of PARIS thrombotic risk score to new patient cohort because of the absence of external validation of this study. Fourth, we used only variables on hospitalization in this study, without taking into consideration changes in medical parameters and post discharge treatment.” (page 37, lines 421 - 428).

Reviewer’s comment

Minor Issues are grammatical in lines 48-53 would suggest to better copyedit this part "...and has developed (into) one of the most popular treatment(s) in modern cardiovascular disease. It is necessary for (would take out "the") patients who under(go) PCI (no plural s required)..." 

Author’s Reply

We apologize for those mistakes and have corrected those mistakes in page 3, lines 52 - 53.

Reviewer’s comment

Lastly also a minor issue in our opinion there is a lack of discussion regarding the findings of cardiac death in the Low, Intermediate, High Risk PARIS score patients as described in Graph B of Figure 1. The focus as it rightly should be on all-cause mortality and CHF events. It would be ideal and interesting to discuss the findings of cardiac death in particular perhaps why these grafts initially partial converge and then after one sees the high risk patients diverge from the intermediate and low risk patients around the ~800 days mark.

Author’s Reply

Thank you for your helpful comment. The 64 cardiac deaths included 48 deaths from heart failure, therefore, the majority of cardiac deaths were heart failure death. Unfortunately, we could not fully explain the reason why the Kaplan-Meier curves for cardiac mortality initially partial converge and the high-risk group diverged from other two groups around 800 days. The Kaplan-Meier curves for hospitalization due to heart failure also initially partial converge and the high-risk group diverged from other two groups around 800 days. The worsening of heart failure with aging may be one of the reasons for the results that you pointed out. We have added the following sentences about these issues in page 35 - 36, lines 381 - 386. 

“The reason why the Kaplan-Meier curves for cardiac mortality and hospitalization due to heart failure partial converge and the high-risk group diverged from other two groups around 800 days is not unclear. One possible reason is the 64 cardiac deaths included 48 deaths from heart failure, therefore, the majority of cardiac deaths were heart failure death. The worsening of heart failure with aging may be one of the reasons for the results.”

Thank you again for your invaluable comments to our manuscript.

Responses to Reviewer #2

We appreciate your detailed reviews on our manuscript. We have revised our manuscript according to your comments. 

Reviewer’s comment

1- Study rationale and hypothesis: The authors chose to calculate the PARIS score for all subjects and present a study to justify its prognostic utility for long term outcomes. While the study population, follow-up duration, comprehensiveness of data and event rate is suitable for such goal, the choice of PARIS score over other available and widely used score for ACS and PCI such as GRACE, TIMI, CADILLAC, among others is not well justified in the introduction or the discussion. The authors highlighted the strengths of their study in comparison to those done by others in the discussion, but it is not true that GRACE and TIMI were not studied for longer term outcomes. Further, the PARIS score albeit simple and easy to use yet is limited with choice of variables (no age, gender, and co-morbidities such as anemia included in the score). Hence, in order to give validity to their study, I most certainly recommend comparing the prognostic value of PARIS score in this population against GRACE and TIMI score and see if it outperforms either one. By doing so, you will give your study more clinical meaning and strength to your chosen score.

Author’s Reply

Thank you for your helpful comment. As you pointed out, the descriptions of conventional risk scores including GRACE, TIMI and CADILLAC were lacking in our original manuscript. According to your suggestion, we have revised the Introduction that firstly described the characteristics of those well-known risk scores, which could estimate the short- to mid-term outcomes of ACS patients (GRACE predicts 6 months mortality, TIMI predicts 14 days and CADILLAC predicts 1 month and 1-year mortalities). To assess the prognostic ability of PARIS thrombotic risk score for predicting mortality, we additionally conducted the receiver operating curve analysis for PARIS thrombotic risk score and GRACE risk score in patients with ACS and compared the areas under the curves (AUC) between two scores, according to your recommendation. The AUC for predicting 6 months mortality was 0.69 in PARIS thrombotic risk score, which was comparable to 0.68 in GRACE score, as described below. Unfortunately, the comparing of PARIS thrombotic risk score to TIMI risk score could not be performed, since the data for calculating TIMI risk score were not available in this cohort. The equivalence of PARIS thrombotic score to GRACE score for predicting mid-term mortality gives validity to the study, therefore we highlighted the long-term prognostic value of PARIS thrombotic risk score for not only all-cause mortality but also cardiac mortality and heart failure event, in addition to the original role for prediction of thrombotic event. We have added the result of ROC analysis in the revised manuscript (page　34, line 336 – 339 and Fig. 4), and described about GRACE, TIMI and CADILLAC in Introduction (page 3, line 55 - 60) and Discussion (page 34 - 35, line 348 - 373). 

“The AUC values of the PARIS thrombotic risk score and the GRACE score for all-cause mortality in 6 months were 0.69 (95% confidence intervals 0.57–0.80) and 0.68 (95% confidence intervals 0.52–0.84). ROC analysis demonstrated that there was no significant difference between these two scores (Fig. 4).” (page 34, line 336 – 339)

Figure 4. ROC to predict all-cause mortality for 6 months in patients with acute coronary syndrome (ACS). 

“Three well-known risk scores are the Global Registry of Acute Coronary Events (GRACE) score [1-3], the Thrombolysis in Myocardial Infarction (TIMI) score[4] and the Controlled Abciximab and Device Investigation to Lower Late Angioplasty Complications (CADILLIAC) score[5]. These scores combine and weigh various predictors to calculate the risk of acute coronary syndrome (ACS) for an individual patient[6].” (page 3, line 55 - 60)

“The risk stratification tools for coronary artery disease after PCIs such as the GRACE score[1-3] ,the TIMI[4] and the CADILLIAC score[5] were reported to have predictive abilities for mortality in patients after PCIs. The GRACE score, which is used for overall risk assessment in patients with ACS, including ST elevation myocardial infarction (STEMI) and non-ST elevation ACS (NSTE-ACS) was designed to calculate the probability of death or myocardial infarction during admission and at 6 months by weighting 8 risk factors: age, heart rate, systolic blood pressure, initial serum creatine, Killip class, hospitalization due to cardiac arrest, positive cardiac biomarker, and ST-segment deviation[1-3, 22]. The TIMI risk score, which is often used for risk assessment in patients with NSTE-ACS, is calculated by using 7 factors: age (≥65 years); at least 3 coronary risk factors (family history, hypertension, dyslipidemia, diabetes mellitus, and smoking); known significant (≥50%) coronary stenosis; ST changes ≥ 0.5 mm on electrocardiogram; at least 2 episodes of angina in 24 hours; aspirin use in the past 7 days; and increased cardiac markers. Most of these factors can be assessed immediately after the admission. As the score increases, the incidence of major cardiovascular complications in the following 2 weeks increases synergistically[4, 22]. The CADILLIAC score, which is often used for risk assessment in patients with STEMI, evaluates several different prognostic variables including LVEF, creatinine clearance, Killip class, final TIMI flow, age, anemia and presence of three vessel disease. Utilizing the CADILLIAC risk score patients were stratified by low (CADILLIAC score 0-2), intermediate (CADILLIAC score 3-5), and high (CADILLIAC score ≥ 6) risk groups. The CADILLIAC score accurately predict 30-day and 1-year mortality after primary PCI[5].” (page 34 - 35, line 348 - 373)

We have cited the following papers as new references (#1-6, 22).

[1] Granger CB, Goldberg RJ, Dabbous O, Pieper KS, Eagle KA, Cannon CP, et al. Predictors of hospital mortality in the global registry of acute coronary events. Arch Intern Med. 2003;163:2345-53.

[2] Eagle KA, Lim MJ, Dabbous OH, Pieper KS, Goldberg RJ, Van de Werf F, et al. A validated prediction model for all forms of acute coronary syndrome: estimating the risk of 6-month postdischarge death in an international registry. Jama. 2004;291:2727-33.

[3] Fox KA, Dabbous OH, Goldberg RJ, Pieper KS, Eagle KA, Van de Werf F, et al. Prediction of risk of death and myocardial infarction in the six months after presentation with acute coronary syndrome: prospective multinational observational study (GRACE). Bmj. 2006;333:1091.

[4] Antman EM, Cohen M, Bernink PJ, McCabe CH, Horacek T, Papuchis G, et al. The TIMI risk score for unstable angina/non-ST elevation MI: A method for prognostication and therapeutic decision making. Jama. 2000;284:835-42.

[5] Halkin A, Singh M, Nikolsky E, Grines CL, Tcheng JE, Garcia E, et al. Prediction of mortality after primary percutaneous coronary intervention for acute myocardial infarction: the CADILLAC risk score. J Am Coll Cardiol. 2005;45:1397-405.

[6] Poldervaart JM, Langedijk M, Backus BE, Dekker IMC, Six AJ, Doevendans PA, et al. Comparison of the GRACE, HEART and TIMI score to predict major adverse cardiac events in chest pain patients at the emergency department. Int J Cardiol. 2017;227:656-61.

[22] Kimura K, Kimura T, Ishihara M, Nakagawa Y, Nakao K, Miyauchi K, et al. JCS 2018 Guideline on Diagnosis and Treatment of Acute Coronary Syndrome. Circ J. 2019;83:1085-196.

Reviewer’s comment

2- Study design and methodology: This is a retrospective cohort study not a prospective one as highlighted in the limitations section. The authors nicely outlined all the steps they undertook to ascertain their outcomes and all the clinical variables of the study, all of which are strengths that can be highlighted in the limitation section.

Recommendations:

- However, there has to be data provided on (1) the type of ACS (STEMI, NSTEMI, UA) and (2) any exclusion criteria (did you include consecutive patients, exclude those with unavailable data on follow-up or STEMI, etc.).

Author’s Reply

Thank you for your helpful comments.　We have showed the type of ACS in Table 1 and added exclusion criteria in page 4, line 88. 

“Exclusion criteria were defined as death at discharge.”

Reviewer’s comment

- Give the presence of competing risk between HF hospitalization and cardiac mortality, the authors are urged to present competing risks survival analysis in their Cox regression for the HF hospitalization and cardiac mortality outcomes.

Author’s Reply

Thank you for pointing out this important issue. We have investigated cardiac mortality and hospitalization due to heart failure by cumulative incidence competing risk method. The cumulative incidence curves have revealed that hospitalization due to heart failure after adjusting competing risks was highest in high-risk group among the three groups as shown Figure 2(B). Furthermore, results from the Fine and Gray model (hazard of the subdistribution model) of association of PARIS thrombosis risk score with cardiac mortality and hospitalization due to heart failure were presented in Tables 6-8. We have added the result of cumulative incidence competing risk method and Fine and Gray model in the revised manuscript (page　15, line 210 – 212 and page 26 – 32, line 276 - 327). 

“In cumulative incidence competing risk method (Fig 2), the hospitalization due to heart failure after adjusting competing risk was highest in high-risk group among the 3 groups (P < 0.001).” (page 15, line 210 – 212)

Figure 2. The cumulative incidence curves for cardiac mortality (A) and hospitalization due to heart failure (B) in the high, intermediate, and low PARIS thrombotic risk score groups.

“Fine and Gray model (hazard of the subdistribution model) 

 The Fine and Gray model (hazard of the subdistribution model) of association of the PARIS thrombotic score, which are presented as categorical variables (groups), with cardiac mortalities and hospitalization due to heart failure are presented in Tables 6–7.”

“Moreover, we performed the Fine and Gray model (hazard of the subdistribution model) of association of the PARIS thrombotic risk score, which are presented as continuous variables (per 1-point increase), with cardiac mortalities and hospitalization due to heart failure as sensitivity analyses (Table 8). In the subdistribution multivariable models, a high PARIS thrombotic score was determined to be an independent predictor of hospitalization due to heart failure after adjusting for other confounding factors (hazard ratio 1.28 per 1-point increase, 95% confidence intervals 1.10–1.50, P = 0.001).” (page 26 – 32, line 276 - 327)

Table 6. Fine and Gray model (hazard of the subdistribution model) for cardiac mortality

Table 7. Fine and Gray model (hazard of the subdistribution model) for hospitalization due to heart failure

Table 8. Fine and Gray model (hazard of the subdistribution model) of PARIS thrombotic score (continuous variables) for cardiac mortalities and hospitalization due to heart failure

Reviewer’s comment

- I recommend to do a stratified analysis on the basis on presentation (ACS vs stable angina) as it is important to see if the PARIS score's prognostic potential holds for either presentation. Comparing the score with GRACE and TIMI is invalid for those with stable angina.

Author’s Reply

Thank you for your important comment. We have conducted subgroup analysis to assess the prognostic value of the PARIS thrombosis risk score for all-cause and cardiac mortality and incidence of hospitalization due to heart failure by clinical presentations. There was no interaction between ACS and stable AP. We have added this issue in page 15 - 16, line 223 – 235 and Figure 3. 

“Furthermore, to assess the prognostic value of the PARIS thrombosis risk score for all-cause and cardiac mortality and incidence of hospitalization due to heart failure by clinical presentations, we conducted subgroup analysis (Fig. 3). Subgroup analysis was performed to compare the high-risk group with the low-risk group. The hazard ratios of the PARIS thrombotic risk score on all-cause mortality in patients presenting stable angina pectoris and acute coronary syndrome were 1.72 (95% confidence intervals 1.01–2.93) and 3.30 (95% confidence intervals 1.63–6.67). The hazard ratios of the PARIS thrombotic risk score on cardiac mortality in patients presenting stable angina pectoris and acute coronary syndrome were 1.55 (95% confidence intervals 0.62–3.84) and 7.92 (95% confidence intervals 1.05–59.80). The hazard ratios of the PARIS thrombotic risk score on the incidence of hospitalization due to heart failure in patients presenting stable angina and acute coronary syndrome were 3.66 (95% confidence intervals 1.69–7.90) and 4.99 (95% confidence intervals 1.53–16.29). There was no interaction.” (page 15 - 16, line 223 – 235)

Figure 3. Forest Plot of hazard ratios by clinical presentations.

Reviewer’s comment

3- Limitations: The biggest limitation of the study is not the small sample size as authors mentioned but (1) retrospective design with inherent limitation to selection and ascertainment biases, (2) the absence of external validation of the proposed score, (3) absence of comparison of PARIS score with other widely used and validated risk scores such as GRACE and TIMI.

Author’s Reply

Thank you for your important comments. We have corrected the limitations as below;

“The present study had several limitations. First, as a retrospective cohort study, the study may be with inherent limitation to selection and ascertainment biases. Second, clinical practices in PCI and medical treatment in the study may have some difference in the setting of each period. Third, it may be difficult to determine the reproducibility and generalizability of PARIS thrombotic risk score to new patient cohort because of the absence of external validation of this study. Fourth, we used only variables on hospitalization in this study, without taking into consideration changes in medical parameters and post discharge treatment.” (page 37, line 421 - 428). 

Regarding the absence of comparison of PARIS thrombotic risk score with other risk score, we have described as above.

Reviewer’s comment

4- Language, technical terms and design: The authors need to improve the writing of the paper in terms of English language, particularly in the introduction. The words "dependable" and "helpful" to describe PCI are not appropriate, and repeating the aims statement with the previous sentence of what's missing in the literature (i.e. the long-term prognostic use of PARIS score) is also not needed. I suggest getting the help of a native English language speaker to proofread the paper.

Further, I could not fully read the tables presented as they were embedded in the text and truncated on the right, making half the tables missing to me. The KM curves are also of very poor quality and need to be presented in a higher resolution.

Author’s Reply

We apologize for mistakes of English language and careless of tables and figures. We have corrected “dependable and helpful” to “well-established” in page 3, line 55. In revised manuscript, you can read fully tables and high-resolution KM curves.

Thank you again for your invaluable comments to our manuscript.

---

## [Decision Letter · Decision Letter 1]

4 Apr 2022

PONE-D-21-38539R1Clinical usefulness of the pattern of non-adherence to anti-platelet regimen in stented patients (PARIS) thrombotic risk score to predict long-term all-cause mortality and heart failure hospitalization after percutaneous coronary intervention.PLOS ONE

Dear Dr. Akama,

Thank you for submitting your manuscript to PLOS ONE. The author's have substantially revised their manuscript and improved the overall quality and impact of the manuscript.  However, several outstanding relatively minor issues need to be addressed before it is suitable for publication in PLOS ONE. Therefore, we invite you to submit a revised version of the manuscript that addresses the points raised during the review process. In addition to the points raised by Reviewer #2, the Editor asks that you review the manuscript for grammatical integrity. For example, there are several places where single sentences serve as stand-alone paragraphs. These should be incorporated into adjacent paragraphs.  

We look forward to receiving your revised manuscript.

Kind regards,

Jeffrey J. Rade, MD

Academic Editor

PLOS ONE

Journal Requirements:

Reviewers' comments:

Reviewer's Responses to Questions

**Comments to the Author**

1. If the authors have adequately addressed your comments raised in a previous round of review and you feel that this manuscript is now acceptable for publication, you may indicate that here to bypass the “Comments to the Author” section, enter your conflict of interest statement in the “Confidential to Editor” section, and submit your "Accept" recommendation.

Reviewer #2: All comments have been addressed

2. Is the manuscript technically sound, and do the data support the conclusions?

Reviewer #2: Yes

3. Has the statistical analysis been performed appropriately and rigorously? 

Reviewer #2: Yes

4. Have the authors made all data underlying the findings in their manuscript fully available?

Reviewer #2: Yes

5. Is the manuscript presented in an intelligible fashion and written in standard English?

Reviewer #2: No

6. Review Comments to the Author

Reviewer #2: The authors did an excellent job responding to all the comments raised, particularly ones related to the scientific rationale and statistical analysis of the data. I have no follow up comments to my previously raised one.

However, I do think that the data presented are elegantly tabulated and charted and deserve a better discussion.

The only major comment I currently have is that there is needs to be major improvement to the discussion section in terms of writeup. It is really hard to follow, with no clear flow of ideas of objective, a lot of redundancy and unnecessary details, and relatively weak critical thinking and reasoning of the data. Here are my suggestions:

1- You need to show why is your data important relative to prior studies (one in particular you cited by Zhao et al.). You need to go in detail about their patient population, how they validated their score against other ones, what makes you study better, etc. (you can't just say longer term follow up).

2- Avoid redundancy. Every paragraphs end by "we show that PARIS score predict outcomes...predict mortality...etc.". You need to put more thought into how your score can be used in the clinical setting (for example, can be used preoperatively, postop to allow for shared decision making and prevent HF hospitalization, etc.).

3- Need to have stronger arguments and better flow of paragraphs. Each paragraph needs to be linked to the one before, start and end with a compelling statement about a certain finding, and end with a concluding statement about the following paragraph or future applications/outlook of your findings.

In summary:

- For each paragraph, identify a study implication about why our study is novel, what it adds to the literature, what it suggestsPARIS score. Always start with a statement about that implication

- Expand on that implication (ie its application) with evidence for or against it, if it exists

- Close with strong statement about what that implication means in the grand scale of things

7. PLOS authors have the option to publish the peer review history of their article (what does this mean?). If published, this will include your full peer review and any attached files.

Reviewer #2: No

---

## [Author Response · Author response to Decision Letter 1]

30 May 2022

Responses to Academic Editor‘s comment

Editor’s comment

In addition to the points raised by Reviewer #2, the Editor asks that you review the manuscript for grammatical integrity. For example, there are several places where single sentences serve as stand-alone paragraphs. These should be incorporated into adjacent paragraphs. 

Author’s Reply

Thank you for your helpful comment. We have found two places where single sentences server as stand-alone paragraphs (page 3, lines 62–64 and page 40, lines 474–475 in the revised manuscript with track changes). We have removed these sentences, as shown below, because they had previously caused some difficulties with regard to following the manuscript. Accordingly, the reference [7] as shown below has been removed. Additionally, we have asked the native English speaker to proofread and have also been modified to make the manuscript easier to understand and to correct any previous grammatical errors as described in “the revised manuscript with track changes”.

“Moreover, it is necessary for patients who undergo PCI or stent implantation to take a dual antiplatelet therapy (DAPT) with aspirin and a P2Y12 receptor inhibitor in order to prevent serious stent-related thrombotic complications after PCIs [7].” (page 3, lines 62–64)

“Findings in this study indicate that the PARIS thrombotic risk score may be used clinically beyond its intended use.” (page 40, lines 474–475)

“[7] Mauri L, Kereiakes DJ, Yeh RW, Driscoll-Shempp P, Cutlip DE, Steg PG, et al. Twelve or 30 months of dual antiplatelet therapy after drug-eluting stents. N Engl J Med. 2014;371:2155-66.“

The following reference has been deleted because the content is supplemented by reference [17].

[18] Ponikowski P, Voors AA, Anker SD, Bueno H, Cleland JGF, Coats AJS, et al. 2016 ESC Guidelines for the diagnosis and treatment of acute and chronic heart failure: The Task Force for the diagnosis and treatment of acute and chronic heart failure of the European Society of Cardiology (ESC)Developed with the special contribution of the Heart Failure Association (HFA) of the ESC. Eur Heart J. 2016;37:2129-200.

Reference [15] has been added to correct reference [16].

[15] Correction: Using standardized serum creatinine values in the Modification of Diet in Renal Disease Study Equation. Ann Intern Med. 2021;174:584.

The following references have been delated as the manuscript has been revised.

[23] Tang EW, Wong CK, Herbison P. Global Registry of Acute Coronary Events (GRACE) hospital discharge risk score accurately predicts long-term mortality post acute coronary syndrome. Am Heart J. 2007;153:29-35.

[24] Valgimigli M, Bueno H, Byrne RA, Collet JP, Costa F, Jeppsson A, et al. 2017 ESC focused update on dual antiplatelet therapy in coronary artery disease developed in collaboration with EACTS: The Task Force for dual antiplatelet therapy in coronary artery disease of the European Society of Cardiology (ESC) and of the European Association for Cardio-Thoracic Surgery (EACTS). Eur Heart J. 2018;39:213-60.

[25] Tanaka S, Sakata R, Marui A, Furukawa Y, Kita T, Kimura T. Predicting long-term mortality after first coronary revascularization: – the Kyoto model –. Circ J. 2012;76:328-34.

[26] Nammas W, Kiviniemi T, Schlitt A, Rubboli A, Valencia J, Lip GYH, et al. Value of DAPT score to predict adverse outcome in patients with atrial fibrillation undergoing percutaneous coronary intervention: A post-hoc analysis from the AFCAS registry. Int J Cardiol. 2018;253:35-9.

Responses to Reviewer #2

We appreciate your detailed review of our manuscript. We believe we have revised the manuscript accordingly. 

Reviewer’s comment

The authors did an excellent job responding to all the comments raised, particularly ones related to the scientific rationale and statistical analysis of the data. I have no follow up comments to my previously raised one.

However, I do think that the data presented are elegantly tabulated and charted and deserve a better discussion.

The only major comment I currently have is that there is needs to be major improvement to the discussion section in terms of writeup. It is really hard to follow, with no clear flow of ideas of objective, a lot of redundancy and unnecessary details, and relatively weak critical thinking and reasoning of the data. Here are my suggestions:

1- You need to show why is your data important relative to prior studies (one in particular you cited by Zhao et al.). You need to go in detail about their patient population, how they validated their score against other ones, what makes you study better, etc. (you can't just say longer term follow up).

Author’s Reply

Thank you for your helpful suggestions. We have described in detail about the differences between our study and the prior one. We have addressed this issue in the Discussion (page 36, line 391–page 37, line 403).

“A few studies have reported on the association between PARIS thrombotic risk score and mortality in patients with PCIs. Zhao et al. reported that the PARIS thrombotic risk score was a significant independent predictor of 2-year mortality in patients after PCIs [26]. One of the advantages of the present study is that our follow-up period was much longer than that in Zhao et al.’s report. Their study population only included patients who had undergone drug eluting stent (DES) implantation, whereas the present study also included patients who had not undergone such implantation, which is useful in terms of stent-less PCI. Moreover, their study did not include a multivariate analysis in assessing the prognostic value of the PARIS thrombotic risk score for all-cause mortality. In contrast, in the present study we performed multivariate Cox proportional hazard analysis followed by competing risk analysis to assess the prognostic value of the PARIS thrombotic risk score. Therefore, the present study provides a new insight into the clinical usefulness of the PARIS thrombotic risk score.” (page 36, line 391–page 37, line 403)

Reviewer’s comment

2- Avoid redundancy. Every paragraph end by "we show that PARIS score predict outcomes...predict mortality...etc.". You need to put more thought into how your score can be used in the clinical setting (for example, can be used preoperatively, postop to allow for shared decision making and prevent HF hospitalization, etc.).

Author’s Reply

According to your comment, we have added the following sentences, which include more insight how the PARIS score can be used in the clinical setting (page 36, lines 383–390).

“Heart failure remains a major cause of mortality, morbidity, hospitalization, and poor quality of life in patients with coronary artery disease [25], thus identifying patients at risk of heart failure is important to improve their prognosis after PCIs. The PARIS thrombotic risk score can be used to allow for shared decision making and prevent hospitalization due to heart failure pre- and post-operatively. We believe that the results of the current study will help clinicians to identify high-risk patients and to plan early examinations and interventions.” (page 36, lines 383–390)

Regarding avoiding redundancy, please see our next reply.

Reviewer’s comment

3- Need to have stronger arguments and better flow of paragraphs. Each paragraph needs to be linked to the one before, start and end with a compelling statement about a certain finding, and end with a concluding statement about the following paragraph or future applications/outlook of your findings.

Author’s Reply

We have reconsidered the flow of paragraphs and reconstructed the discussion. At first, we modified “However, these risk scores were estimated for predicting short to mid-term prognosis, thus the long-term follow-up after PCI were poorly documented[23]. Therefore, the present study has the advantages of enrolling all patients who underwent PCI and may aid in predicting long-term prognosis after PCI.” (page 37, lines 395–399 in the revised manuscript with track changes) to “However, there have been few risk scores that include patients with not only ACS, but also chronic coronary syndrome, and can be used to predict long term mortalities. The ability of PARIS thrombotic risk score to predict the 6-month mortality of patients with ACS was comparable to that of the GRACE score; hence, we examined the long-term prognostic value of PARIS thrombotic risk score for both ACS and chronic coronary syndrome” (page 35, lines 368–373). 

Moreover, the following sentences have been removed to avoid redundancy and be linked to the one before, start and end with a compelling statement about a certain finding, and end with a concluding statement about the following paragraph or future applications/outlook of our findings.

“The results of this study … improve the prognosis after PCI.” (page 38, line 405–page 39, line 432 in the revised manuscript with track changes)

“Findings in this study indicate that the PARIS thrombotic risk score may be used clinically beyond its intended use.” (page 40, line 474–page 41, line 475 in the revised manuscript with track changes)

Moreover, we have added the sentences below to achieve the same purpose.

“Heart failure remains a major cause of mortality, morbidity, hospitalization, and poor quality of life in patients with coronary artery disease [25]; thus, identifying patients at risk of heart failure is important to improve their prognosis after PCIs. The PARIS thrombotic risk score can be used to allow for shared decision making and prevent hospitalization due to heart failure pre- and postoperatively. We believe that the results of the current study will help clinicians to identify high-risk patients and to plan early examinations and interventions.” (page 36, lines 383–390)

The text below has been repositioned on page 37, lines 411–416 to improve the flow of paragraphs.

“The reason why the Kaplan-Meier curves for cardiac mortality and hospitalization due to heart failure partially converge and the high-risk group diverged from those in the other two groups at around 800 days is unclear. One possible reason is that the 64 cardiac deaths included 48 deaths from heart failure; therefore, the majority of cardiac deaths were due to heart failure. The worsening of heart failure with age may be one of the reasons for these results.” (page 37, lines 411–416)

Accordingly, the following references have been removed.

[23] Tang EW, Wong CK, Herbison P. Global Registry of Acute Coronary Events (GRACE) hospital discharge risk score accurately predicts long-term mortality post acute coronary syndrome. Am Heart J. 2007;153:29-35.

[24] Valgimigli M, Bueno H, Byrne RA, Collet JP, Costa F, Jeppsson A, et al. 2017 ESC focused update on dual antiplatelet therapy in coronary artery disease developed in collaboration with EACTS: The Task Force for dual antiplatelet therapy in coronary artery disease of the European Society of Cardiology (ESC) and of the European Association for Cardio-Thoracic Surgery (EACTS). Eur Heart J. 2018;39:213-60.

[25] Tanaka S, Sakata R, Marui A, Furukawa Y, Kita T, Kimura T. Predicting long-term mortality after first coronary revascularization: – the Kyoto model –. Circ J. 2012;76:328-34.

[26] Nammas W, Kiviniemi T, Schlitt A, Rubboli A, Valencia J, Lip GYH, et al. Value of DAPT score to predict adverse outcome in patients with atrial fibrillation undergoing percutaneous coronary intervention: A post-hoc analysis from the AFCAS registry. Int J Cardiol. 2018;253:35-9.

We have asked the native English speaker to proofread and have also been modified to make the manuscript easier to understand and to correct any previous grammatical errors as described in “the revised manuscript with track changes”.

In summary: 

- For each paragraph, identify a study implication about why our study is novel, what it adds to the literature, what it suggests PARIS score. Always start with a statement about that implication

- Expand on that implication (ie its application) with evidence for or against it, if it exists

- Close with strong statement about what that implication means in the grand scale of things

Author’s Reply

Thank you for putting together your comments. Please see our responses to your comments. Details are given above.

---

## [Decision Letter · Decision Letter 2]

11 Jul 2022

PONE-D-21-38539R2Clinical usefulness of the pattern of non-adherence to anti-platelet regimen in stented patients (PARIS) thrombotic risk score to predict long-term all-cause mortality and heart failure hospitalization after percutaneous coronary intervention.PLOS ONE

Dear Dr. Akama,

Thank you for submitting your manuscript to PLOS ONE. After careful consideration, we feel that it has merit but does not fully meet PLOS ONE’s publication criteria as it currently stands. Therefore, we invite you to submit a revised version of the manuscript that addresses the points raised during the review process.

We look forward to receiving your revised manuscript.

Kind regards,

Chiara Lazzeri

Academic Editor

PLOS ONE

Reviewers' comments:

Reviewer's Responses to Questions

**Comments to the Author**

1. If the authors have adequately addressed your comments raised in a previous round of review and you feel that this manuscript is now acceptable for publication, you may indicate that here to bypass the “Comments to the Author” section, enter your conflict of interest statement in the “Confidential to Editor” section, and submit your "Accept" recommendation.

Reviewer #2: All comments have been addressed

Reviewer #3: (No Response)

2. Is the manuscript technically sound, and do the data support the conclusions?

Reviewer #2: Yes

Reviewer #3: Yes

3. Has the statistical analysis been performed appropriately and rigorously? 

Reviewer #2: Yes

Reviewer #3: Yes

4. Have the authors made all data underlying the findings in their manuscript fully available?

Reviewer #2: No

Reviewer #3: Yes

5. Is the manuscript presented in an intelligible fashion and written in standard English?

Reviewer #2: No

Reviewer #3: Yes

6. Review Comments to the Author

Reviewer #2: Thanks for your attempt at fixing the grammar and Discussion section with the help of native English speaker. Unfortunately, I still don't think it is a good discussion that conveys the importance of your findings. I think you need to spend more time showing the feasibility of using your score as it only includes clinical characteristics and remove all the unnecessary details of what the GRACE, TIMI and CADILLAC scores contain (basically remove the second paragraph).

You also need to remove strong conclusions such as this :"The PARIS thrombotic risk score can be used to allow for shared decision making and prevent hospitalization due to heart failure pre- and postoperatively. We believe that the results of the current study will help clinicians to identify high-risk patients and to plan early examinations and interventions." & "Therefore, the present study provides a new insight into the clinical

403 usefulness of the PARIS thrombotic risk score" You need to tone down these concluding statements.

You need to spend most of the discussion highlighting the importance and feasibility of your score like you did in the third paragraph and end of second paragraph. The connection and flow of ideas between paragraphs need to improved as the last 3 paragraphs tackle completely unrelated ideas. I suggest you show what the score predict (mortality, HF, etc) and why is that important as other scores don't predict all of these, then move to discuss how feasible it is to use the score, and then write on the limitations. Remove paragraph with lines 404-410 as it doesn't add much to the discussion. Keep it simple and straightforward.

Reviewer #3: Authors tried to address all the previously raised points.

This Reviewer would like to perform the following further comments and criticisms:

1. Page 4. Study protocol. Overall, 1061 PCI patients were enrolled during a period longer than 8 years. The only exclusion criterium reported in the manuscript was death during hospitalization. On average, about 130 PCI patients per year were enrolled. This mean that the enrolling PCI center was a very low volume hospital (not aligned to the current international standards) and/or in-hospital mortality was very high and/or other exclusion criteria were adopted but not reported. Please review this major issue.

2. Page 5. Study protocol. How was considered deaths from unknow cause? How many deaths from unknown cause were reported?

3. Page 6. Co-morbidities. Dyslipidemia definition is not in line with current guidelines and therapeutic cut-offs. Please review.

4. Table 1. The use of coronary imaging devices was extremely high (97.2%). Was the study database based on a registry dedicated intracoronary imaging modalities during PCI?

5. Tables. The number of Tables and Figures is an issue, and it should be reduced.

6. Page 36 and 37. Discussion. Please better explain how the PARIS “thrombotic” risk score can predict both the deaths and the hospitalizations for heart failure, the later not always related to a thrombotic event (i.e., but to poor drug adherence). In addition, absence of data regarding long-term pharmacological and intervention treatments should be better discussed in the study limitation section.

7. PLOS authors have the option to publish the peer review history of their article (what does this mean?). If published, this will include your full peer review and any attached files.

Reviewer #2: No

Reviewer #3: No

---

## [Author Response · Author response to Decision Letter 2]

21 Aug 2022

Responses to Reviewer #2 

We appreciate your detailed review of our manuscript (PONE-D-21-38539). We have revised our manuscript according to your comments.

Reviewer’s comment 1

Thanks for your attempt at fixing the grammar and Discussion section with the help of native English speaker. Unfortunately, I still don't think it is a good discussion that conveys the importance of your findings. I think you need to spend more time showing the feasibility of using your score as it only includes clinical characteristics and remove all the unnecessary details of what the GRACE, TIMI and CADILLAC scores contain (basically remove the second paragraph). You also need to remove strong conclusions such as this: "The PARIS thrombotic risk score can be used to allow for shared decision making and prevent hospitalization due to heart failure pre- and postoperatively. We believe that the results of the current study will help clinicians to identify high-risk patients and to plan early examinations and interventions." & "Therefore, the present study provides a new insight into the clinical usefulness of the PARIS thrombotic risk score" You need to tone down these concluding statements. 

Author’s Reply 1

Thank you for your helpful comment. In response to your indication, we have removed the second paragraph of the Discussion section which described details of GRACE, TIMI and CADILLAC scores. Below is the second paragraph that has been removed. 

“The GRACE score, which is used for overall risk assessment in patients with ACS, including ST elevation myocardial infarction (STEMI) and non-ST elevation ACS (NSTE-ACS), was designed to calculate the probability of death or myocardial infarction at admission and at six months after undergoing PCI by weighting eight risk factors: age, heart rate, systolic blood pressure, initial serum creatine, Killip class, hospitalization due to cardiac arrest, positive cardiac biomarker, and ST-segment deviation [1-3, 21]. The TIMI risk score, which is often used for risk assessment in patients with NSTE-ACS, is calculated using seven factors: age (≥ 65 years); at least three coronary risk factors (family history, hypertension, dyslipidemia, diabetes mellitus, and smoking); known significant coronary stenosis (≥ 50%); ST changes ≥ 0.5 mm on electrocardiogram; at least two episodes of angina within the previous 24 hours, aspirin use in the past 7 days; and increased cardiac markers. Most of these factors can be assessed immediately after admission. As the score increases, the risk of major cardiovascular complications in the following 2 weeks increases synergistically [4, 21]. The CADILLAC score, which is often used for risk assessment in patients with STEMI, evaluates several different prognostic variables, including LVEF, creatinine clearance, Killip class, final TIMI flow, age, anemia and presence of three-vessel disease. Utilizing the CADILLAC risk score, the patients were stratified into low (CADILLAC score 0–2), intermediate (score 3–5), and high (score ≥ 6) risk groups. The CADILLAC score can be used to accurately predict 30-day and 1-year mortality after primary PCI [5].” 

Also, in response to your comment that the tone of the results is too strong, we have revised the sentences in the Discussion section (page 35, lines 366–371 and page 36, lines 383–384), toning them down. The following sentences are the revised sentences.

“Present study has shown the possibility that physicians are able to apply the PARIS thrombotic score not only to estimate the risk of coronary thrombotic events, but also to assess the risk of death and worsening heart failure for patients after PCIs. The PARIS thrombotic risk score may be able to serve as a clinical tool for shared decision making to prevent thrombotic event, hospitalization due to heart failure and death in the clinical practice post PCI.” (page 35, lines 366–371)

“Therefore, the present study suggests that it may provide a new insight into the clinical usefulness of the PARIS thrombotic risk score.” (page 36, lines 383–384)

Reviewer’s comment 2

You need to spend most of the discussion highlighting the importance and feasibility of your score like you did in the third paragraph and end of second paragraph. The connection and flow of ideas between paragraphs need to improve as the last 3 paragraphs tackle completely unrelated ideas. I suggest you show what the score predict (mortality, HF, etc) and why is that important as other scores don't predict all of these, then move to discuss how feasible it is to use the score, and then write on the limitations. Remove paragraph with lines 404-410 as it doesn't add much to the discussion. Keep it simple and straightforward.

Author’s Reply 2

Thank you for your helpful comments and suggestions. In line with your suggestion, we highlighted that the PARIS thrombotic score in the present analysis included not only ACS patients but also patients with chronic coronary syndrome, and could predict the outcomes of death and heart failure hospitalization, which other risk scores do not. We have revised the sentences as below (page 34, line 350 – page 35, line 358).

“The ability of PARIS thrombotic risk score to predict the 6-month mortality of patients with ACS was comparable to that of the GRACE score; hence, we examined the long-term prognostic value of PARIS thrombotic risk score for both ACS and chronic coronary syndrome. In addition, we have shown that the PARIS thrombotic risk score can predict events of heart failure hospitalization that the above scores do not. Heart failure remains a major cause of mortality, morbidity, hospitalization, and poor quality of life in patients with coronary artery disease [21]; thus, identifying patients at risk of heart failure is important to improve their prognosis after PCIs.” (page 34, line 350 – page 35, line 358)

As you pointed out, the last 3 paragraphs of the Discussion section are irrelevant. According to your suggestion, the following 2 paragraph has been removed to keep the discussion simple and straightforward. 

“In patients with coronary artery disease after PCIs, various prognostic factors on mortality have been reported previously [27-29]. Co-morbidities such as diabetes mellitus [30], history of acute coronary syndrome, current smoking [31], CKD [32], and history of PCI or CABG [33, 34] have already reported to be risk factors for poor prognosis in patients with coronary artery disease. Since these factors are also components of the PARIS thrombotic risk score, it is possible that this score could be used to predict all-cause death or hospitalization due to heart failure.” 

“The reason why the Kaplan-Meier curves for cardiac mortality and hospitalization due to heart failure partially converge and the high-risk group diverged from those in the other two groups at around 800 days is unclear. One possible reason is that the 64 cardiac deaths included 48 deaths from heart failure; therefore, the majority of cardiac deaths were due to heart failure. The worsening of heart failure with age may be one of the reasons for these results.” 

We thank you again for your valuable comments on our manuscript.

Responses to Reviewer #3 

We appreciate your detailed review of our manuscript (PONE-D-21-38539). We have revised our manuscript according to your comments.

Reviewer’s comment 1

Page 4. Study protocol. Overall, 1061 PCI patients were enrolled during a period longer than 8 years. The only exclusion criterium reported in the manuscript was death during hospitalization. On average, about 130 PCI patients per year were enrolled. This mean that the enrolling PCI center was a very low volume hospital (not aligned to the current international standards) and/or in-hospital mortality was very high and/or other exclusion criteria were adopted but not reported. Please review this major issue.

Author’s Reply 1

Thank you for your helpful comment. The present study has enrolled only patients who have undergone their initial PCI at our hospital. We apologize that we missed to write down about this indication. While, our hospital performed about 230 cases of PCI annually between 2010 and 2018 including target lesion or target vessel revascularization and staged PCI after initial one. We think this meets the criteria for adequate institutional volume achieving good outcomes, which is defined as over 200 PCIs per year (J Am Coll Cardiol. 2013; 62: 357-96). We have revised the manuscript about inclusion criteria in the Method section (page 4, lines 79–82). The below is the revised sentences.

“This was an observational study, which enrolled a total of 1,061 consecutive patients who underwent their initial PCI due to stable angina pectoris or acute coronary syndrome (ACS) at Fukushima Medical University Hospital between January 2010 and May 2018.” (page 4, lines 79–82)

Reviewer’s comment 2

Page 5. Study protocol. How was considered deaths from unknow cause? How many deaths from unknown cause were reported?

Author’s Reply 2

Thank you for your helpful comment. Deaths from unknow cause were counted as all cause death. 5 deaths from unknown cause have been reported.

Reviewer’s comment 3

Page 6. Co-morbidities. Dyslipidemia definition is not in line with current guidelines and therapeutic cut-offs. Please review.

Author’s Reply 3

Thank you for your valuable comment. As you mentioned, the definition of dyslipidemia used in the present study is not in line with current guidelines, since we do not know the initial cholesterol levels of patients who were on statins at the time of index PCI. Hence, we inevitably defined dyslipidemia as the recent use of cholesterol-lowering drugs, a triglyceride value of ≥ 150 mg/dL, a low-density lipoprotein (LDL) cholesterol value of ≥ 140 mg/dL, and/or a high-density lipoprotein (HDL) cholesterol value of < 40 mg/dL, as described in the Method section.

Reviewer’s comment 4

Table 1. The use of coronary imaging devices was extremely high (97.2%). Was the study database based on a registry dedicated intracoronary imaging modalities during PCI?

Author’s Reply 4

Thank you for your important comment. This study is not based on any registry dedicated intracoronary imaging modalities. In Japan, intracoronary imaging is accepted in insured medical treatment, so that it is quite usual to use intracoronary imaging devices during daily PCI procedures. Therefore, the use of coronary imaging devices was extremely high.

Reviewer’s comment 5

Tables. The number of Tables and Figures is an issue, and it should be reduced.

Author’s Reply 5

Thank you for your valuable suggestion. As you noted, the number of tables has increased from the first draft. However, due to the meaningful peer review so far, detailed analysis has been performed, and the number of figures and tables has increased. There is no mention of the number of figures and tables in the submission instructions, and we would like to keep the number of figures and tables as they are for better understanding.

Reviewer’s comment 6

Page 36 and 37. Discussion. Please better explain how the PARIS “thrombotic” risk score can predict both the deaths and the hospitalizations for heart failure, the later not always related to a thrombotic event (i.e., but to poor drug adherence). In addition, absence of data regarding long-term pharmacological and intervention treatments should be better discussed in the study limitation section.

Author’s Reply 6

Thank you for your important suggestion. The possible reason why the PARIS thrombotic risk score can also predict all-cause mortality and heart failure due to hospitalization is that major predictors used in the PARIS thrombotic risk score were associated with the risk factors for a poor prognosis of coronary artery disease. It has already been reported that history of acute coronary syndrome and chronic kidney disease is also a risk factor for hospitalization due to heart failure. These explanations were firstly described in the Discussion section, however, the Reviewer 2 recommended us that the above content should not be included in the manuscript because it complicates the content of the discussion section. 

While, as you mentioned, the achievement rate of guideline directed medical therapy and the presence or absence of any interventional treatment to the coronary artery disease and the heart failure may affect thrombotic events, death and heart failure hospitalization. We have revised the manuscript and described about this point in the Limitation section (page 36, lines 387–391). Following are the added sentences.

“Second, the clinical practices regarding medical treatment and interventional treatment in the present study may have some difference in the setting of each period. Especially, the achievement rate of guideline directed medical therapy for coronary artery disease and the presence or absence of interventional treatment may affect the rate of thrombotic events, death and heart failure hospitalization after PCI.” (page 36, lines 387–391)

We thank you again for your valuable comments on our manuscript.

---

## [Editor Report · Decision Letter 3]

25 Aug 2022

Clinical usefulness of the pattern of non-adherence to anti-platelet regimen in stented patients (PARIS) thrombotic risk score to predict long-term all-cause mortality and heart failure hospitalization after percutaneous coronary intervention.

PONE-D-21-38539R3

Dear Dr. Akama,

We’re pleased to inform you that your manuscript has been judged scientifically suitable for publication and will be formally accepted for publication once it meets all outstanding technical requirements.

Kind regards,

Chiara Lazzeri

Academic Editor

PLOS ONE
---

## [Editor Report · Acceptance letter]

5 Sep 2022

PONE-D-21-38539R3 

Clinical usefulness of the pattern of non-adherence to anti-platelet regimen in stented patients (PARIS) thrombotic risk score to predict long-term all-cause mortality and heart failure hospitalization after percutaneous coronary intervention. 

Dear Dr. Akama:

I'm pleased to inform you that your manuscript has been deemed suitable for publication in PLOS ONE. Congratulations! Your manuscript is now with our production department. 

Kind regards, 

on behalf of

Dr. Chiara Lazzeri 

Academic Editor

PLOS ONE